# Logistic Regression Through the Veil of Imprecise Data

## Abstract

Logistic regression is a popular supervised learning algorithm used to assess the probability of a variable having a binary label based on some predictive features. Standard methods can only deal with precisely known data; however, many datasets have interval uncertainties due to measurement error, censoring or missing data that traditional methods either reduce to a single point or completely disregard. This paper shows that it is possible to include these intervals by considering an imprecise logistic regression model using the set of possible models obtained from values within the intervals. This method can express the epistemic uncertainty neglected by traditional methods.

## 1 Introduction

Logistic regression is used to predict the probability of a binary outcome as a function of some predictive variable. In medicine, for example, logistic regression can be used to predict the probability of an individual having a disease where the values of risk factors are known. While logistic regression is most commonly used for binary outcomes, multinomial logistic regression extends this into events with any number of labels (Menard, 2010, Chapter 1). However, many decisions and events are binary (yes/no, passed/failed, alive/dead, etc.), and for the sake of simplicity, we will restrict our discussion and examples to binary outcome logistic regression. Additionally, unlike discriminant function analysis, logistic regression does not require predictor variables to be normally distributed, linearly related or have equal variance (Press & Wilson, 1978).

There are many practical applications for logistic regression across many different fields. For example, in the medical domain, risk factors in the form of continuous data–such as age–or categorical data–such as gender–may be used to fit a model to predict the probability of a patient surviving an operation (Bagley et al., 2001; Neary et al., 2003). In engineering systems, logistic regression can be used to determine whether a mineshaft is safe (Palei & Das, 2009); to predict the risk of lightning strikes (Lambert & Wheeler, 2005) or landslides (Ohlmacher & Davis, 2003). In the arts, it can explore how education impacts museum attendance or watching a performing arts performance (Kracman, 1996). It is also possible to predict the probability of sports results using logistic regression (Li et al., 2021). Due to its wide range of applications, logistic regression is considered a fundamental machine learning algorithm with many modern programming languages having packages for users to experiment with, such as *Scikit-learn* (Pedregosa et al., 2011) in Python, which has been used for the analysis within this paper.

Traditionally it has been assumed that all of the values of the features and labels used in logistic regression are *precisely* known. However, in practice, there can be considerable imprecision in the features and labels used in the regression analysis and the application of the regression model. Analyses using data from combined studies with inconsistent measurement methods can even result in datasets with varying degrees of uncertainty. Likewise, the outcome data can be uncertain if there is ambiguity in the classification scheme (good/bad). However, even relatively straightforward classifications (alive/dead) can yield uncertainty when a subject leaves a study, and the outcome is unknown. Within statistics, censored data is sometimes of this form Rabinowjtz et al. (1995); Lindsey & Ryan (1998). In the case of continuous variables, the interval reflects the measurement uncertainty, while in the binary outcome, the interval is the vacuous $[0, 1]$ because the correct classification is unclear.

There are multiple methods of dealing with interval data with the features of a logistic regression model. The problem may also be considered a subproblem within symbolic data analysis (Billard & Diday, 2003; Whitaker et al., 2021; Beranger et al., 2022). These methods often require approximations to be made to simplify the process and allow the use of standard logistic regression techniques. The most straightforward approach is to discard the interval data, assuming that the epistemic uncertainty that the intervals represent is small compared to the sampling uncertainty or natural variability in the data or that values are missing at random or missing completely at random (Rubin, 1976; Ferson et al., 2007; Kreinovich, 2009; Hosmer Jr et al., 2013). These assumptions are likely to be untenable in practice. Another approach is to treat interval data as uniform distribution based on the "equidistribution hypothesis" that assumes each possible value to be equally likely; thus, the intervals are modelled as a uniform distribution (Bertrand, 2000; Billard & Diday, 2000; Bock & Diday, 2000; Beranger et al., 2022). This idea has its roots in the principle of insufficient reason, first described by both Bernoulli and Laplace, and more recently known as the principle of indifference (Keynes, 1921). Alternatively, the interval is commonly represented by the interval's midpoint, which represents the mean and median of a uniform distribution or a random value from within the interval (Osler et al., 2010). While these approaches are computationally expedient, as will be shown in Section 3.3, they underrepresent the imprecision by presenting a single middle-of-the-road logistic regression.

Similar methods include performing a conjoint logistic regression using the interval endpoints or averaging separate regressions performed on the endpoints of the intervals (de Souza et al., 2008; 2011). While these various methods make different assumptions about the data within the interval ranges, ultimately, they still transform interval data such that the final results can be represented by a single binary logistic regression (de Souza et al., 2011).

The approach proposed within this paper for dealing with interval data in logistic regressions is based on imprecise probabilities and considers the set of models rather than a single one (Walley, 1991; Manski, 2003; Ferson et al., 2007; Nguyen et al., 2012). This is similar to approaches proposed for dealing with interval uncertainty within linear regression (Utkin & Coolen, 2011; Wiencierz, 2013; Schollmeyer, 2021; Tretiak et al., 2022). These approaches, alongside other methods for interval linear regression (Gioia et al., 2005; Fagundes et al., 2013) are not directly translatable to logistic regression as they require the use of least squares methods ill-suited to dichotomous problems (Hosmer Jr et al., 2013).

If separate logistic regressions are generated via maximum likelihood estimation from the interval data and displayed graphically, the envelope of these models can be considered as an imprecise logistic regression model. The 'true' model–the model that would have been fitted if there was no epistemic uncertainty associated with the sample–would always be contained within the bounds of the best possible imprecise model. The primary benefit of such an approach is that it represents the epistemic uncertainty removed by traditional methods. Additionally, this method can also handle the case of intervals in discrete risk factors. This imprecise approach makes the fewest assumptions but can be computationally challenging for large datasets (Ferson et al., 2002; 2007; Kreinovich, 2009).

In the case of uncertainty in the outcome status used within logistic regression, traditionally, there is little that can be done but to discard these data points as they cannot be used as part of the analysis or to use a semi-supervised learning methodology (Amini & Gallinari, 2002; Chapelle et al., 2006; Chi et al., 2019). However, the proposed imprecise logistic regression technique can be used to include unlabelled examples within the dataset. Again the imprecise approach does not require making the assumptions required by other methods to fit a model.

This paper continues as follows: in Section 2, precise logistic regression is reviewed, Section 3 and Section 4 introduce the imprecise logistic regression for data with intervals within the features and labels respectively. In both these sections, a 1-dimensional synthetic dataset is used to demonstrate the methodology before it is compared to alternative methods on both the synthetic data and a real-world example. The purpose of the comparison with existing techniques is not to . Finally, in Section 5 the method is used on a dataset which contains intervals in both the features and labels and is contrasted against a dataset from the literature.

## 2 Precise Logistic Regression

Let $\mathbf{x}$ be a $m$-dimensional covariate with a binary label, $y \in \{1, 0\}$. Logistic regression can be used to model the probability that $y = 1$ using:

$$\Pr(y = 1|\mathbf{x}) = \pi(\mathbf{x}) = \frac{1}{1 + \exp\left(-\left(\beta_0 + \beta_1 x_1 + \cdots + \beta_m x_m\right)\right)} \tag{1}$$

where $\beta_0, \beta_1, \ldots$ are a set of unknown regression coefficients. If dataset $D$ contains $n$ samples

$$D = \left\{ \begin{array}{c} \left((x_1^{(1)} x_2^{(1)} \cdots x_m^{(1)}), y_1\right) \\ \left((x_1^{(2)} x_2^{(2)} \cdots x_m^{(2)}), y_2\right) \\ \vdots \\ \left((x_1^{(n)} x_2^{(n)} \cdots x_m^{(n)}), y_n\right) \end{array} \right\}, \tag{2}$$

a logistic regression model can be trained on $D$, $\mathcal{LR}(D)$, by finding optimal values of $\beta_0, \beta_1, \ldots, \beta_m$ for the observed data. This is often done using *maximum likelihood estimation* (Menard, 2010; Myung, 2003), although other techniques exist, for instance through Bayesian analysis Jaakkola (1997); O'Brien & Dunson (2004).

A classification, $\hat{y}$, can be made from the logistic regression model by selecting a threshold value, $C$, and then defining

$$\hat{y} = \begin{cases} 1 & \text{if } \pi(\mathbf{x}) \geq C \\ 0 & \text{if } \pi(\mathbf{x}) < C \end{cases} \tag{3}$$

The simplest case is when $C = 0.5$, implying $\hat{y}$ is more likely to be true than false. However, this value could be different depending on the use of the model and the risk appetite of the analyst. For example, in medicine, a small threshold value may be used in order to produce a conservative classification and therefore reduce the number of false negative results. Where predictions are made within this paper, $C = 0.5$ unless otherwise stated.

### 2.1 Demonstration

To demonstrate, a synthetic 1-dimensional dataset ($D_1$) with a sample size of fifty was used to train a logistic regression model, $\mathcal{LR}(D_1)$, as shown in Figure 1. After training the model, it is useful to ask the question "how good is the model?" For logistic regression there are several ways in which that can be done, see Hosmer Jr et al. (2013, pp. 157–169) or Kleinbaum & Klein (2010, pp.318–326). For the analysis in this paper, we will consider the receiver operating characteristic graph and area under curve statistic, discriminatory performance visualisations (Royston & Altman, 2010) as well as the sensitivity and specificity of the classifications made by the algorithm.

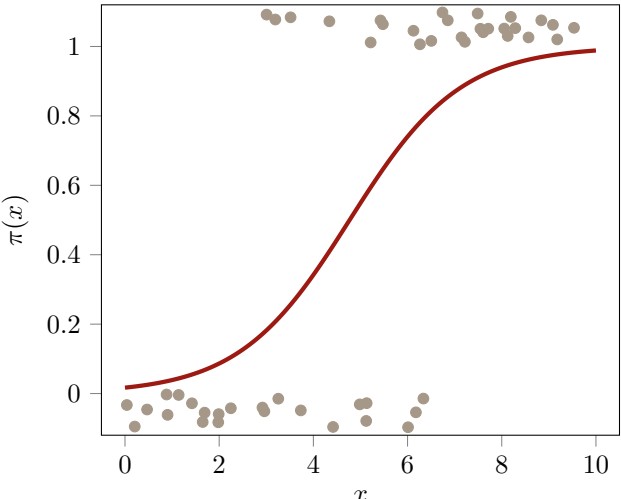

Figure 1: Logistic regression curve, $\mathcal{LR}(D_1)$, created by fitting the model on dataset $D_1$ (shown with scattered points).

Royston & Altman (2010) introduced visualisations to assess the discriminatory performance of the model by considering a scatter plot of the true outcome (jittered for clarity) vs the estimated probability. Such a plot is shown in Figure 2a. A perfectly discriminating model would have two singularities with all the points with outcome = 1 at (1,1) and all the points with outcome = 0 at (0,0). In general, the better the classifier, the more clustered the points would be towards these values, with the points on the upper band having larger probabilities and the points on the lower band having lower probabilities. From Figure 2a we can see that there is significant clustering towards the endpoints, showing that the model has excellent discriminatory performance.

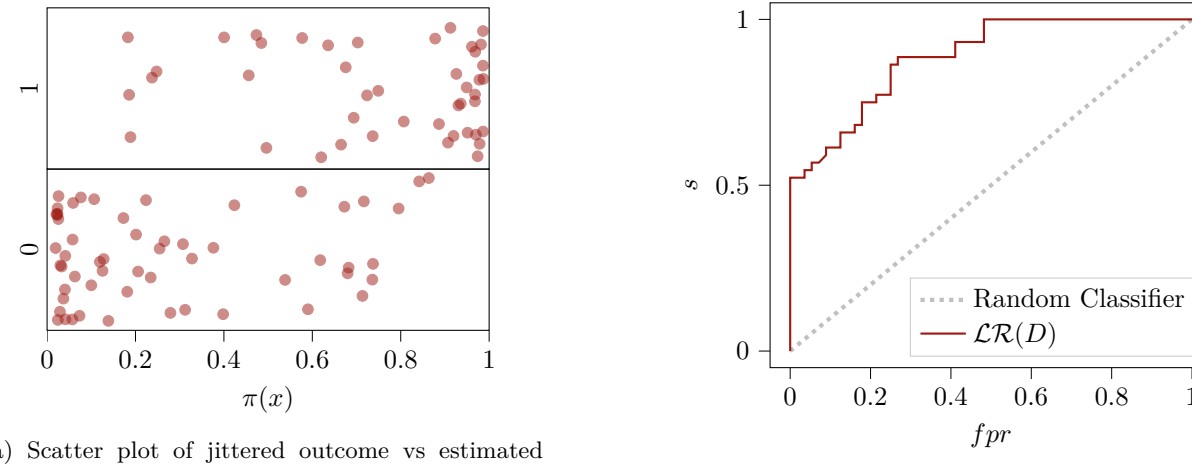

(a) Scatter plot of jittered outcome vs estimated probability.

(b) ROC curve.

Figure 2: Two plots to show the discriminatory performance of the logistic regression model shown in Figure 1.

We can make and compare the predictions made from the logistic regression model using a larger dataset that has been generated using the same method as above. Tabulating these results in a confusion matrix for the base predictions is shown in Table 1.

| | 1 | 0 | Total |
|---|---|---|---|
| Predicted 1 | 34 | 14 | 48 |
| Predicted 0 | 10 | 42 | 52 |
| Total | 44 | 56 | 100 |

Table 1: Confusion matrix for 100 test data points using predictions from $\mathcal{LR}(X)$ shown in Figure 1.

There are numerous statistics than can be derived from confusion matrices to express the performance of classifiers. For this analysis, we shall consider the sensitivity $s$, the fraction of positive individuals correctly identified as such and the specificity $t$, the fraction of negative individuals correctly identified. Mathematically

$$s = \frac{\text{True Positive}}{\text{Total Number of Positives}} \tag{4}$$

and

$$t = \frac{\text{True Negative}}{\text{Total Number of Negatives}}. \tag{5}$$

From Table 1 we can calculate that $s = 0.773$ and $t = 0.750$. As confusion matrices and statistics are calculated from them depending on the cutoff value chosen ($C$ from Equation 3), a complete way of determining the classification performance of models is by considering the receiver operating characteristic (ROC) curve of the model (Kleinbaum & Klein, 2010; Hosmer Jr et al., 2013). ROC curves can be compared graphically and by considering the area under the curve (AUC). The better the model is, the closer the AUC would be to 1. The worst possible AUC would be 0.5, as again, anything lower than that would be improved by simply switching the classification. For the ROC curve shown in Figure 2b, AUC = 0.887.

## 3  Uncertainty in Features

If there is interval uncertainty within dataset $D$,

$$D = \left\{ \begin{array}{c} \left( \left( \left[ \underline{x_1^{(1)}}, \overline{x_1^{(1)}} \right] \left[ \underline{x_2^{(1)}}, \overline{x_2^{(1)}} \right] \cdots \left[ \underline{x_m^{(1)}}, \overline{x_m^{(1)}} \right] \right), y_1 \right) \\ \left( \left( \left[ \underline{x_1^{(2)}}, \overline{x_1^{(2)}} \right] \left[ \underline{x_2^{(2)}}, \overline{x_2^{(2)}} \right] \cdots \left[ \underline{x_m^{(2)}}, \overline{x_m^{(2)}} \right] \right), y_2 \right) \\ \vdots \\ \left( \left( \left[ \underline{x_1^{(n)}}, \overline{x_1^{(n)}} \right] \left[ \underline{x_2^{(n)}}, \overline{x_2^{(n)}} \right] \cdots \left[ \underline{x_m^{(n)}}, \overline{x_m^{(n)}} \right] \right), y_n \right) \end{array} \right\}, \tag{6}$$

and we have no more information about the true value, $x_j^{(i)\dagger}$, nor are willing to make further assumptions about the true value, only that $x_j^{(i)\dagger} \in \left[ \underline{x_j^{(i)}}, \overline{x_j^{(i)}} \right]$, then it is only possible to partially identify an imprecise logistic regression model for the data, $\mathcal{ILR}(D)$:

$$\mathcal{ILR}(D) = \left\{ \mathcal{LR}\left(D'\right) : \forall D' \in \left\{ \left\{ \begin{array}{c} \left( \left( x_1'^{(1)} \cdots x_1'^{(m)} \right), y_1 \right) \\ \vdots \\ \left( \left( x_n'^{(1)} \cdots x_n'^{(m)} \right), y_n \right) \end{array} \right\} \forall x_j'^{(i)} \in \left[ \underline{x_j^{(i)}}, \overline{x_j^{(i)}} \right] \right\} \right\} \tag{7}$$

i.e. $\mathcal{ILR}(D)$ is the set of all possible logistic regression models that can be created from all possible datasets that can be constructed from the interval data; this ensures that the true logistic regression model, $\mathcal{LR}^{\dagger}$, is contained within the set. This set is infinitely large for continuous data, so estimates must be used to find the set.

Predictions can be made by sampling all the possible models that are contained within $\mathcal{ILR}(D)$ and creating an interval containing the maximum and minimum values, $\pi(\mathbf{x}) = \left[ \underline{\pi(\mathbf{x})}, \overline{\pi(\mathbf{x})} \right]$. When calculating the

probability of a value being 1 under the imprecise model, there is an interval probability $\left[\underline{\pi(x)}, \overline{\pi(x)}\right]$, where $\overline{\pi(x)}$ and $\underline{\pi(x)}$ are the maximum and minimum values of $\pi(x)$ calculated from models within the set. As such, when using the model to perform classifications, this interval means that Equation 3 becomes

$$
\hat{y} = \begin{cases} 1 & \text{if } \underline{\pi}(\mathbf{x}) > C \\ 0 & \text{if } \overline{\pi}(\mathbf{x}) < C \\ [0,1] & \text{if } C\left[\underline{\pi(\mathbf{x})}, \overline{\pi(\mathbf{x})}\right] \end{cases}. \tag{8}
$$

The final line of this equation returns the *dunno* interval, meaning there is uncertainty in determining whether the datum should be predicted 0 or 1, and the model should therefore abstain from providing a classification. It is left up to the analyst to decide what to do with such a result.

Under such a scenario, there are two approaches to characterising the classifier using a confusion matrix. The first is to consider the intervals directly within the confusion matrix. Calculating statistics derived from the confusion matrix requires careful handling for interval calculations (Gray et al., 2022).

To demonstrate, let $a$ be the number of true positives, $b$ be the number of false positives, $c$ be the number of false negatives, and $d$ be the number of true negatives, where $a$, $b$, $c$ and $d$ are all intervals. Since the number of positive individuals is fixed, $a$ and $c$ are oppositely dependent on each other (as $a \to \overline{a}$ then $c \to \underline{c}$), as are $b$ and $d$. These dependencies imply that care must be taken when calculating sensitivity and specificity. Sensitivity needs to be calculated as

$$
s = \frac{a}{\underline{a} + \overline{c}}, \tag{9}
$$

and specificity as

$$
t = \frac{d}{\underline{b} + \overline{d}}. \tag{10}
$$

In this instance, the $s$ and $t$ would themselves both be intervals ($s = [\underline{s}, \overline{s}]$ and $t = [\underline{t}, \overline{t}]$). $\overline{s}$ and $\overline{t}$ would be the sensitivity and specificity of a system that first perfectly classified all the uncertain predictions.

Of course, analysts could use various alternative statistics to describe the classifier's performance, some of which require special care when using imprecise numbers. For instance, *precision* and *recall* are often used within the machine learning literature to assess classifiers. Precision is the fraction of positive predictions that are true positives,

$$
\text{precision} = \frac{\text{True Positive}}{\text{Total Number of Positive Predictions}}, \tag{11}
$$

and recall is analogous to sensitivity. Often quoted alongside these statistics is the $F_1$ score, which is the harmonic mean of these values,

$$
F_1 = 2\frac{\text{precision} \cdot \text{recall}}{\text{precision} + \text{recall}}. \tag{12}
$$

With intervals, precision needs to be calculated using a single-use expression to ensure that there is no artifactual uncertainty within the calculation

$$
\text{precision} = \frac{1}{1 + \frac{b}{a}} \tag{13}
$$

and recall calculated using Equation 9.

The $F_1$ score is again best calculated through a single-use expression

$$
F_1 = 2\frac{1}{\frac{TP+b}{a} + 1} \tag{14}
$$

where $TP$ is the total number of positive predictions ($TP = \underline{a} + \overline{c}$).

An alternative approach to constructing a confusion matrix with uncertain predictions is to tabulate the dunno predictions in a separate row. If the model returned $u$ true positives, $v$ false positives, $w$ false negatives

and $x$ true negatives but did not make a prediction for $y$ positives and $z$ negatives, then the confusion matrix shown in Table 2 can be created.

From this confusion matrix, some useful statistics can be calculated to account for the uncertainty produced by these uncertain classifications. The traditional definitions of sensitivity and specificity can be re-imagined by defining what the *predictive sensitivity s'* as the sensitivity out of the points for which a prediction was made

$$s' = \frac{u}{u + w} \tag{15}$$

and similarly, the *predictive specificity t'* as the specificity for which a prediction was made

$$t' = \frac{x}{v + x}. \tag{16}$$

|              | 1     | 0     | Total     |
|--------------|-------|-------|-----------|
| Predicted 1  | $u$   | $v$   | $P_+$     |
| Predicted 0  | $w$   | $x$   | $P_-$     |
| No Prediction| $y$   | $z$   | $P_\times$|
| Total        | $T_+$ | $T_-$ | $N$       |

Table 2: Alternative confusions matrix where uncertain predictions are tabulated separately.

Two other statistics are useful to describe the data in Table 2. We can define the *positive incertitude $\sigma$* to be the fraction of positive cases for which the model could not make a prediction

$$\sigma = \frac{y}{u + w + y}. \tag{17}$$

Similarly, the *negative incertitude $\tau$* can be defined as the total number of negative cases for which the model could not make a prediction

$$\tau = \frac{z}{v + x + z}. \tag{18}$$

### 3.1 Approaches to Estimating Equation 7

As the set described in Equation 7 is infinitely large approaches need to be taken to estimate it. Since any analysis of $\mathcal{ILR}(D)$ only requires knowledge of the maximum and minimum $\pi(\mathbf{x})$ values ($\left[\underline{\pi(\mathbf{x})}, \overline{\pi(\mathbf{x})}\right]$), the best possible model would only need to contain the models that produce models that for some $\mathbf{x}$ produce either endpoint of the interval.

### 3.1.1 Systematic Search

The ideal method of estimating the set is to search values from within the intervals systematically. This approach requires specifying $P$ as the number of steps within the intervals, then producing a logistic regression model as shown in Algorithm F0. This approach would have complexity $\mathcal{O}(N^P)$ where $P$ is the number of intervals within the set. So whilst it would be the method most accurate to the best-possible method of computing $\mathcal{ILR}(D)$, for large datasets, this algorithm would be exceedingly time expensive and thus impractical.

---

**Algorithm F0:** Systematic search to find $\mathcal{ILR}(D)$ if $D$ has interval uncertainty within its features.

---

**Input:** $D = \left\{ \left( \left( \left[ \underline{x_j^{(i)}}, \overline{x_j^{(i)}} \right] \ \forall i = 0, \ldots, m \right), y_j \right) \ \forall j = 0, \ldots, n \right\}$, $Q$ steps

$\mathcal{ILR}(D) \leftarrow \{\}$;

$\Delta \leftarrow \left\{ \frac{i-1}{N-1} \ \forall i = 1, \ldots, N \right\}$;

**for all combinations** $\left\{ \delta_j^{(i)} \right\} \in \Delta$ **do**

   $D' \leftarrow D$;

   **for all** $i \in \{0, \ldots, m\}$ **do**

      **for all** $j \in \{0, \ldots, n\}$ **do**

         $D_j'^{(i)} \leftarrow \left\{ \underline{x_j^{(i)}} + \delta_j^{(i)} \left( \overline{x_j^{(i)}} - \underline{x_j^{(i)}} \right) \right\}$

      **end**

      1

   **end**

   $\mathcal{ILR}(D) \leftarrow \mathcal{ILR}(D) \cup \{\mathcal{LR}(D')\}$;

**end**

**Output:** $\mathcal{ILR}(D)$

---

### 3.1.2 Method of Minimum and Maximum Coefficients

Since it is only necessary to find the logistic regression models that correspond to extreme values of $\pi(\mathbf{x})$ for all $\pi(\mathbf{x})$. It is possible to reduce the number of models that need to be contained within $\mathcal{ILR}(D)$ to only those that make up the envelope of the set. These lines can be estimated as

$$\mathcal{ILR}(D) = \left\{ \mathcal{LR}\left( D'_{\underline{\beta_i}} \right), \mathcal{LR}\left( D'_{\overline{\beta_i}} \right) \ \forall i = 0, 1, \ldots, m \right\}$$
$$\cup \left\{ \mathcal{LR}\left( \underline{D} \right), \mathcal{LR}\left( \overline{D} \right) \right\} \tag{19}$$

where $D'_{\underline{\beta_0}}$ is the dataset constructed from points within the intervals such that the value of $\beta_0$ is minimised, $D'_{\overline{\beta_0}}$ is the dataset constructed from points within the intervals such that the value of $\beta_0$ is maximised, and so on. $\underline{D}$ corresponds to the dataset created by taking the lower bound of every interval within $D$, similarly for $\overline{D}$. For a dataset with $m$ features, there are $2m + 2^m$ models that are needed to find the bounds of the set. Algorithm F1 can be used to find this set. An illustration of the validity of this estimation can be found in Appendix A. Figure 3 shows all six lines for the six intervals shown compared to the systematic method, with the black dashed lines representing the envelope of the set.

To measure how good an estimate the method is, we can let $A$ be the area of the systematic bounds that are outside the bounds produced by the proposed algorithm as a fraction of the total area found systematically. Ideally, we would see $A = 0$, implying that the set perfectly covered the systematic bounds. The envelope of these models shown withn Figure 3 has $A = 0.0160$.

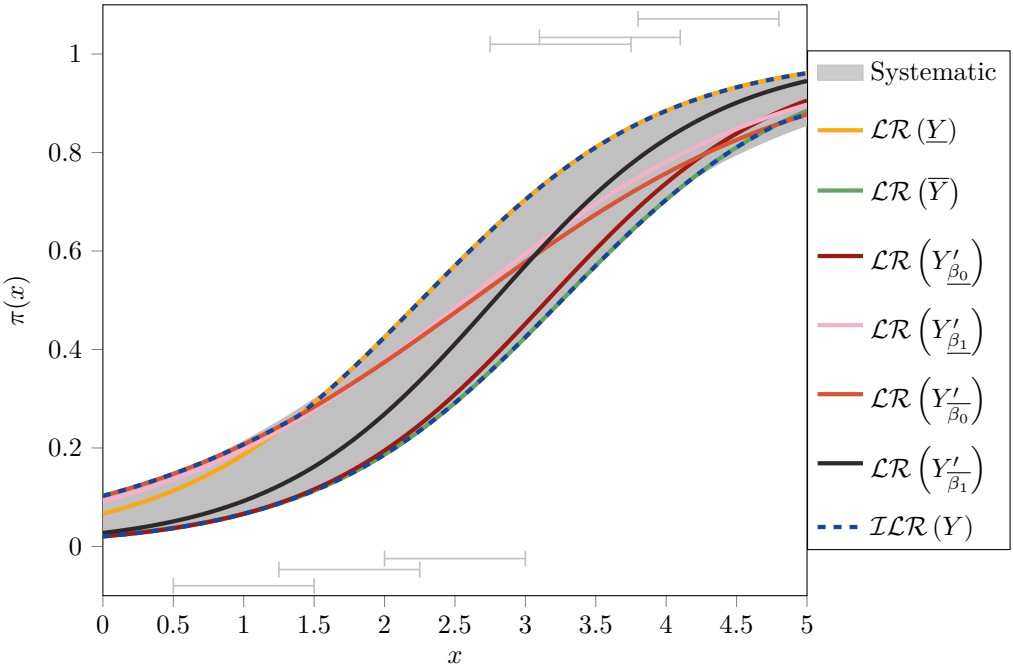

Figure 3: Comparison of systematic approach against Algorithm F1.

**Algorithm F1:** Method to find $\mathcal{ILR}\left(D\right)$ if $D$ has interval uncertainty within its features by calculating the minimum and maximum intercept and coefficients.

**Input:** $D = \left\{ \left( \left( \left[ \underline{x_j^{(i)}}, \overline{x_j^{(i)}} \right] \ \forall i = 0, \ldots, m \right), y_j \right) \ \forall j = 0, \ldots, n \right\}$

$\underline{D} = \left\{ \left( \left( \underline{x_j^{(i)}} \ \forall i = 0, \ldots, m \right), y_j \right) \ \forall j = 0, \ldots, n \right\};$

$\overline{D} = \left\{ \left( \left( \overline{x_j^{(i)}} \ \forall i = 0, \ldots, m \right), y_j \right) \ \forall j = 0, \ldots, n \right\};$

$\mathcal{ILR}\left(D\right) \leftarrow \left\{ \mathcal{LR}\left(\underline{D}\right), \mathcal{LR}\left(\overline{D}\right) \right\};$

**for** $i \leftarrow 0$ **to** $m$ **do**

    Using stochastic optimisation find $D'_{\underline{\beta_i}}$ such that $\mathcal{LR}\left(D'_{\underline{\beta_i}}\right)$ has the minimum value of $\beta_i$;

    $\mathcal{ILR}\left(D\right) \leftarrow \mathcal{ILR}\left(D\right) \cup \left\{ \mathcal{LR}\left(D'\right) \right\};$

    Using stochastic optimisation find $D'_{\overline{\beta_i}}$ such that $\mathcal{LR}\left(D'_{\overline{\beta_i}}\right)$ has the maximum value of $\beta_i$;

    $\mathcal{ILR}\left(D\right) \leftarrow \mathcal{ILR}\left(D\right) \cup \left\{ \mathcal{LR}\left(D'\right) \right\};$

**end**

**Output:** $\mathcal{ILR}\left(D\right)$

### 3.1.3 Minimum and Maximum Spread Approach

If the dataset contains a large number of intervals, then, due to the increasing complexity of the optimisation, Algorithm F1 may take a prohibitively long time to compute. In such a situation, Algorithm F2 can be used to find the imprecise model. This algorithm uses the heuristic that the extreme bounds are likely to be associated with the minimum and maximum spread of points around particular values, a discussion of this heuristic can be found in Appendix B. The complexity of this approach is $\mathcal{O}(2(1 + P))$

The bounds produced by this method for the same six intervals from Figure 3 are shown in Figure 4. This plot has $A = 0.0351$ compared to the systematic search method.

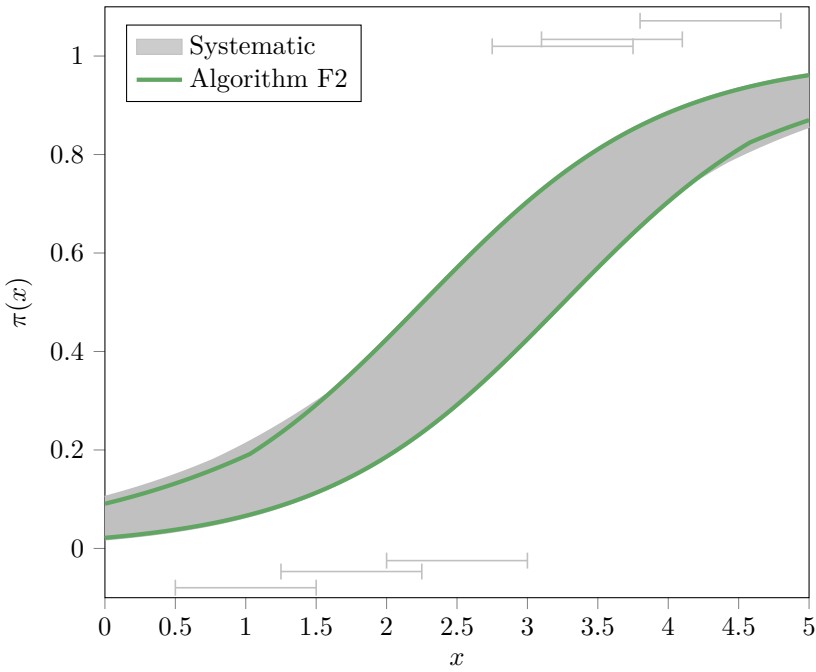

Figure 4: Comparison of systematic approach against Algorithm F2 with $P = 100$.

---

**Algorithm F2:** Minimum and Maximum spread approach to estimate $\mathcal{ILR}\left(D\right)$ if $D$ has interval uncertainty within its features. The procedures for finding the minimum and maximum spread of points around a particular value can be found in Prodcedures 1 and 2 in Appendix B

---

**Input:** $D = \left\{ \left( \left( \left[ \underline{x_j^{(i)}}, \overline{x_j^{(i)}} \right] \ \forall i = 0, \ldots, m \right), y_j \right) \ \forall j = 0, \ldots, n \right\}$

$\underline{D} = \left\{ \left( \left( \underline{x_j^{(i)}} \ \forall i = 0, \ldots, m \right), y_j \right) \ \forall j = 0, \ldots, n \right\};$

$\overline{D} = \left\{ \left( \left( \overline{x_j^{(i)}} \ \forall i = 0, \ldots, m \right), y_j \right) \ \forall j = 0, \ldots, n \right\};$

$\mathcal{ILR}\left(D\right) \leftarrow \left\{ \mathcal{LR}\left(\underline{D}\right), \mathcal{LR}\left(\overline{D}\right) \right\};$

**for all** $p \in \left\{ \frac{k}{P+1} \ \forall k = 1, \ldots, P \right\}$ **do**

    Find $\underline{T}$ such that $\pi_{\mathcal{LR}(\underline{D})}(T) = P;$

    Find $\overline{T}$ such that $\pi_{\mathcal{LR}(\overline{D})}(T) = P;$

    $d_{min} \leftarrow$ minimum spread of points around $\underline{T};$

    $d_{max} \leftarrow$ maximum spread of points around $\overline{T};$

    $\mathcal{ILR}\left(D\right) \leftarrow \mathcal{ILR}\left(D\right) \cup \left\{ \mathcal{LR}\left(d_{min}\right), \mathcal{LR}\left(d_{max}\right) \right\};$

**end**

**Output:** $\mathcal{ILR}\left(D\right)$

---

### 3.1.4 Monte Carlo

A straightforward and computationally expedient way of estimating is to use Monte Carlo to find valid models. This approach, as shown in Algorithm F3, requires the analyst to specify a desired number of iterations, then, for every iteration, make a new dataset by sampling a random value from all intervals within the dataset, then fitting a model on this new dataset. Often this sampling assumes the equidistribution hypothesis and thus models the intervals as a uniform distribution. This approach is likely to underestimate the bounds as Monte Carlo methods are unlikely to find the extreme models (Ferson, 1996).

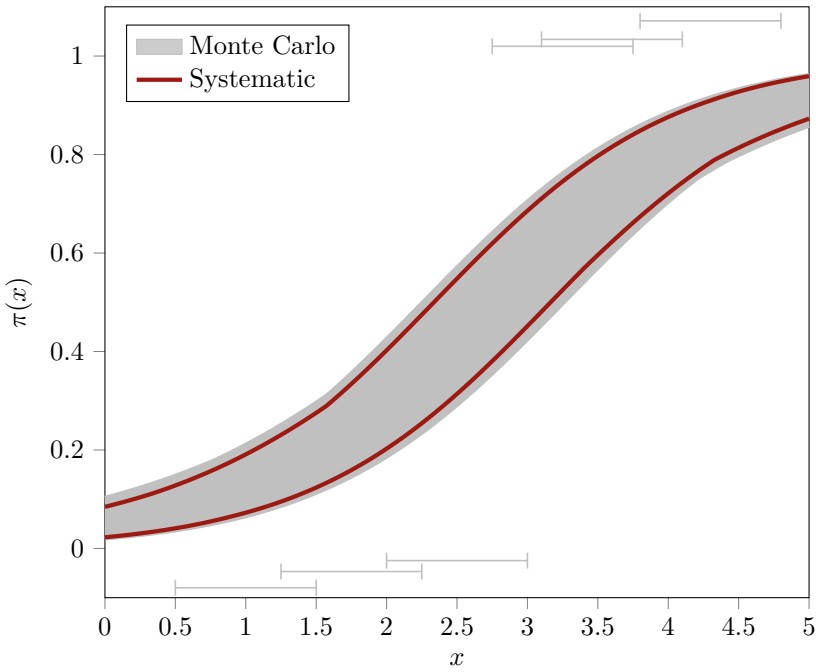

Figure 5: Comparison of systematic approach against a Monte Carlo sample with $10^6$ iterations.

Figure 5 shows a comparison between a Monte Carlo search with $10^6$ iterations and a systematic search for the six intervals shown. As expected, the bounds from the Monte Carlo method fail to enclose the total area revealed by the systematic search. In this case, $A = 0.166$. A higher number of iterations would have led to a smaller $A$ value.

---

**Algorithm F3:** Monte Carlo search to find $\mathcal{ILR}(D)$ if $D$ has interval uncertainty within its features.

---

**Input:** $D = \left\{ \left( \left( \left[ \underline{x_j^{(i)}}, \overline{x_j^{(i)}} \right] \ \forall i = 0, \ldots, m \right), y_j \right) \ \forall j = 0, \ldots, n \right\}$, $P$ steps

$\mathcal{ILR}(D) \leftarrow \{\}$;

**for** $P$ *iterations* **do**

$\quad D' \leftarrow D$;

$\quad D_j'^{(i)} \leftarrow$ random value in $\left[ \underline{x_j^{(i)}}, \overline{x_j^{(i)}} \right]$;

$\quad \mathcal{ILR}(D) \leftarrow \mathcal{ILR}(D) \cup \{\mathcal{LR}(D')\}$;

**end**

**Output:** $\mathcal{ILR}(D)$

---

### 3.2 Alternative Methods

Several authors have suggested different approaches to compute logistic regression models with interval uncertainty. This section will consider three methods to compare with the methods presented in this paper.

#### 3.2.1 Midpoint

The most straightforward approach to dealing with interval data is to produce a precise dataset by replacing the intervals with their midpoints and then fitting a dataset with this midpoint data. i.e.

$$D_m = \left\{ \left( \left( \frac{\underline{x_j^{(i)}} + \overline{x_j^{(i)}}}{2} \right), y_j \right) \ \forall \left[ \underline{x_j^{(i)}}, \overline{x_j^{(i)}} \right] \in D \right\} \tag{20}$$

then

$$\mathcal{LR}_m(D) = \mathcal{LR}(D_m).\tag{21}$$

This dataset can then be used as a precise logistic regression model described in Section 2.

### 3.2.2 de Souza

de Souza et al. (2008; 2011) introduce several methods for characterising the uncertainty with interval features. They conclude that the best method is to perform two separate logistic regressions on the lower and upper bounds of the intervals and average the posterior probabilities to obtain a pooled posterior probability. They find

$$\mathcal{LR}_{dS}(D) = \left\{\mathcal{LR}(\underline{D}), \mathcal{LR}(\overline{D})\right\}\tag{22}$$

and then reduces this to a single logistic regression model based on the average of the outputted probabilities.

$$\pi_{\mathcal{LR}_{dS}(D)}(\mathbf{x}) = \frac{\pi_{\mathcal{LR}(\underline{D})}(\mathbf{x}) + \pi_{\mathcal{LR}(\overline{D})}(\mathbf{x})}{2}.\tag{23}$$

### 3.2.3 Billard-Diday

Billard & Diday (2000) propose a method, based upon Bertrand (2000), for characterising interval uncertainties within linear regression that can be easily extended to logistic regression. Their method assumes that each value from within the interval is equally likely, and therefore constructs the logistic regression models as the uniform mixture of $N$ logistic regression models that are fitted from random samples,

$$\mathcal{LR}_{BD}(D) = \left\{\mathcal{LR}(D_k) : D_k = \left\{\left\{\left((r_j^{(i)}), y_j\right)\right\} \ r_j'^{(i)} \in \left[\underline{x_j^{(i)}}, \overline{x_j^{(i)}}\right]\right\} \ k = 0, \dots, N\right\}\tag{24}$$

like de Souza, they then average the probability from all models

$$\pi_{\mathcal{LR}_{BD}(D)}(\mathbf{x}) = \frac{1}{N} \sum_{\forall l \in \mathcal{LR}_{BD}(D)} \pi_l(\mathbf{x}).\tag{25}$$

This method is computationally the same as Algorithm F3 but takes the average of the found models to produce a precise final model instead of taking the envelope to produce an imprecise model.

### 3.3 Comparison of Methods

Dataset $D_1$ from 2 has been intervalised into dataset $D_2$ using the following transformation $x_i' = [m - \epsilon, m + \epsilon]$ where $m$ is a number drawn from the triangular distribution $\mathrm{T}(x_i - \epsilon, x_i + \frac{\epsilon}{6}, x_i + \epsilon)$ with $\epsilon = 0.375$ for all $x_i \in X$. With this dataset we can use Algorithms F3, F1 and F2 to construct $\mathcal{ILR}(Y)$, as is shown in Figure 6. It would have been too computationally expensive to perform a systematic search using Algorithm F0.

Figure 6 shows that Algorithm F1 produces the most comprehensive bounds and is, therefore, most likely to represent the entire set described by Equation 7. The Monte Carlo approach in Algorithm F3 produces the narrowest bounds with Algorithm F2

For comparison, logistic regression models have been fitted using the methods described in Section 3.2; these models are shown in Figure 7. Whilst there are subtle differences between the different approaches, it is clear that they are all approximately equal. This equivelence is unsurprising as they all implicitly make the equidistribution assumption that a uniform distribution can represent the intervals.

It is also possible to consider situations where data is intervalised differently. Figure 8 shows four different intervalisations of dataset $X$ as described in Table 3. In plot (a), the intervalisation has occurred by placing the true value at the left edge of the interval; similarly, in (b), the true value is at the right edge of the interval. In (c), the value of $x$ impacts the interval's width, and in (d), the label impacts the intervalisation. In this figure, imprecise models have been fitted on the datasets using Algorithms F3, F1 and F2 alongside

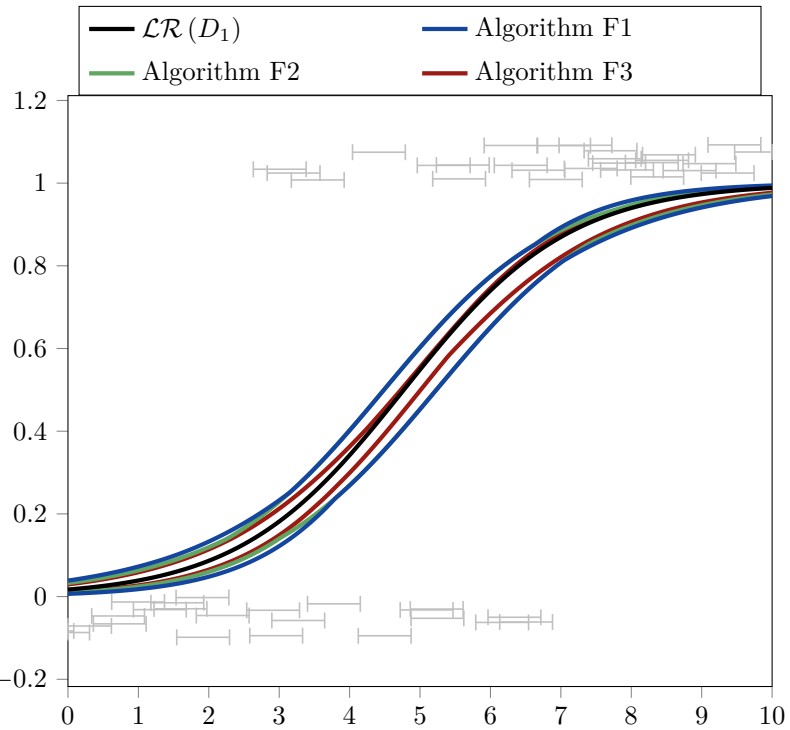

Figure 6: Imprecise logistic regression models fitted using Algorithms F3 (with $10^4$ iterations), F1 and F2 for the interval data (jittered for clarity). $\mathcal{LR}\,(D_1)$ represents the 'true' model from Figure 1.

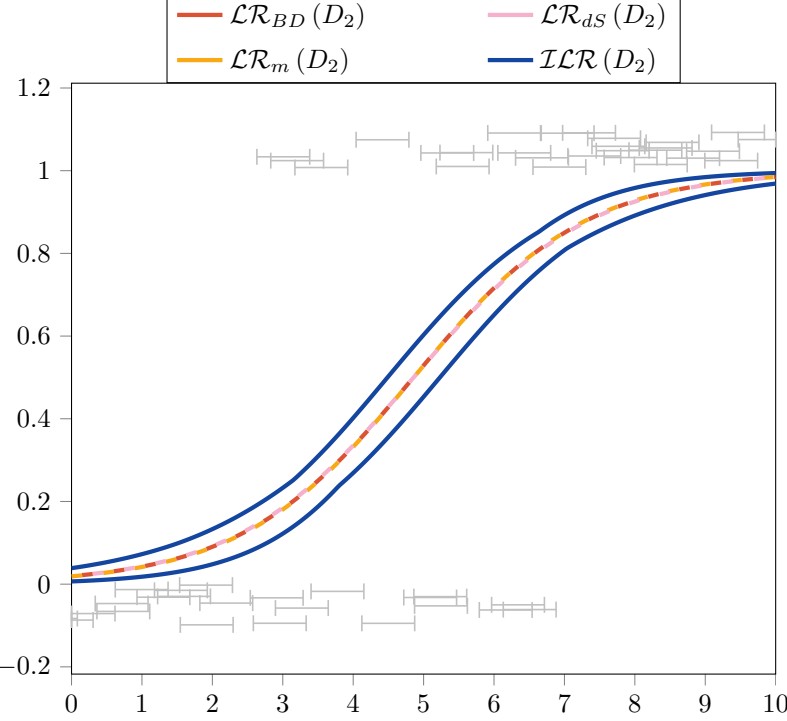

Figure 7: Imprecise logistic regression model fitted using Algorithm F1 and logistic regression models fitted using the alternative methods in Section 3.2 for the interval dataset (jittered for clarity).

the 'true' model (from Figure 1) and the midpoint model. The de Souza and Billard-Diday methods are not shown due to their similarity with the midpoint model.

Looking at all these figures, we see that the imprecise model produced by Algorithms F1 and F2 always bounds the base model. As a result, any interval regression analysis performed would be guaranteed to bound the true model. The model produced by Algorithm F3 fails to do so in (a), (b) and (d). The fact that in (a) and (b), the models produced by Algorithm F1 and Algorithm F2

The figure also shows that there can be significant differences between the base and midpoint models. The alternative approaches provide a good approximation of the true value if the equidistribution hypothesis can be justified, as in plot (c). If the intervalisation depends on the outcome, then the approaches appear inadequate, as is shown in plot (d). This implies that the alternative approaches, and Algorithm F3, should only be used in cases within which one can assume that the data has been intervalised independent of either the true underlying value or the outcome status and each value within the interval is equally likely.

In many real-world datasets, the assumptions that the alternative methods rely on fail to hold, and in those scenarios, only the complete imprecise method would guarantee coverage of the true model.

### 3.4   Red Wine Example

In order to demonstrate the methodology on a real-world dataset, we can use the red wine dataset from Cortez et al. (2009). This dataset contains 11 covariates[1] that can be used to predict the quality of the wine sample based on a scale from 0 to 10. In order to provide a binary classification, we define wine as good if it has quality $\geq 6$. The dataset contains 1599 samples, of which 855 have been classified as good wine. The dataset with added uncertainties is denoted as $R$.

In order to fit the logistic regression model, the dataset has been split into training and test subsets containing half the data in each. To intervalise the data, values have been turned into intervals based on the number of significant figures within the model (for example, 0.5 would become $[0.45, 0.55]$). An imprecise model can then be fitted on the dataset using Algorithm F2.

It is helpful to consider visualisations when discussing the classifier's performance (Figure 9). The simplest of these is the scatter plots shown in Figure 9a. From the plot, most of the wines rated as good were given a high $\pi$ value, and vice versa for the bad wines–although no wine was given a very low $pi$. There is, however, substantial overlap between the two groups. The plot also shows that some wines have a wide interval $\pi$. The plot also shows the size of the intervals for $\pi$. In this instance, all intervals are reasonably consistent and not overly wide.

We can also construct ROC plots and calculate their AUCs, as shown in Figure 9b. In this plot, we can see that the $\mathcal{LR}\left(R^{\dagger}\right)$ and $\mathcal{LR}_m\left(R\right)$ curves are only subtly distinguishable from each other. The same is also true of $\mathcal{LR}_{dS}\left(R\right)$ and $\mathcal{LR}_{BD}\left(R\right)$. The imprecise model bounds all these models. Additionally, for the imprecise model, we can plot a ROC curve for when the model abstains from making a prediction. In this instance $s'$ is plotted against $fpr'(= 1 - t')$. This ROC curve outperforms the others.

The AUC for the curves are $AUC\left[\mathcal{ILR}\left(R\right)\right] = [0.742, 0.872]$, $AUC\left[\mathcal{ILR}\left(R\right)(\,Abstain)\right] = 0.834$, $AUC[\mathcal{LR}\left(R^{\dagger}\right)] \approx AUC\left[\mathcal{LR}_m\left(R\right)\right] = 0.818$ and $AUC[\mathcal{LR}_{dS}\left(R\right)] \approx AUC[\mathcal{LR}_{BD}\left(R\right)] = 0.813$.

---

[1]fixed acidity, volatile acidity, citric acid, residual sugar, chlorides, free sulfur dioxide, total sulfur dioxide, density, pH, sulphates, alcohol

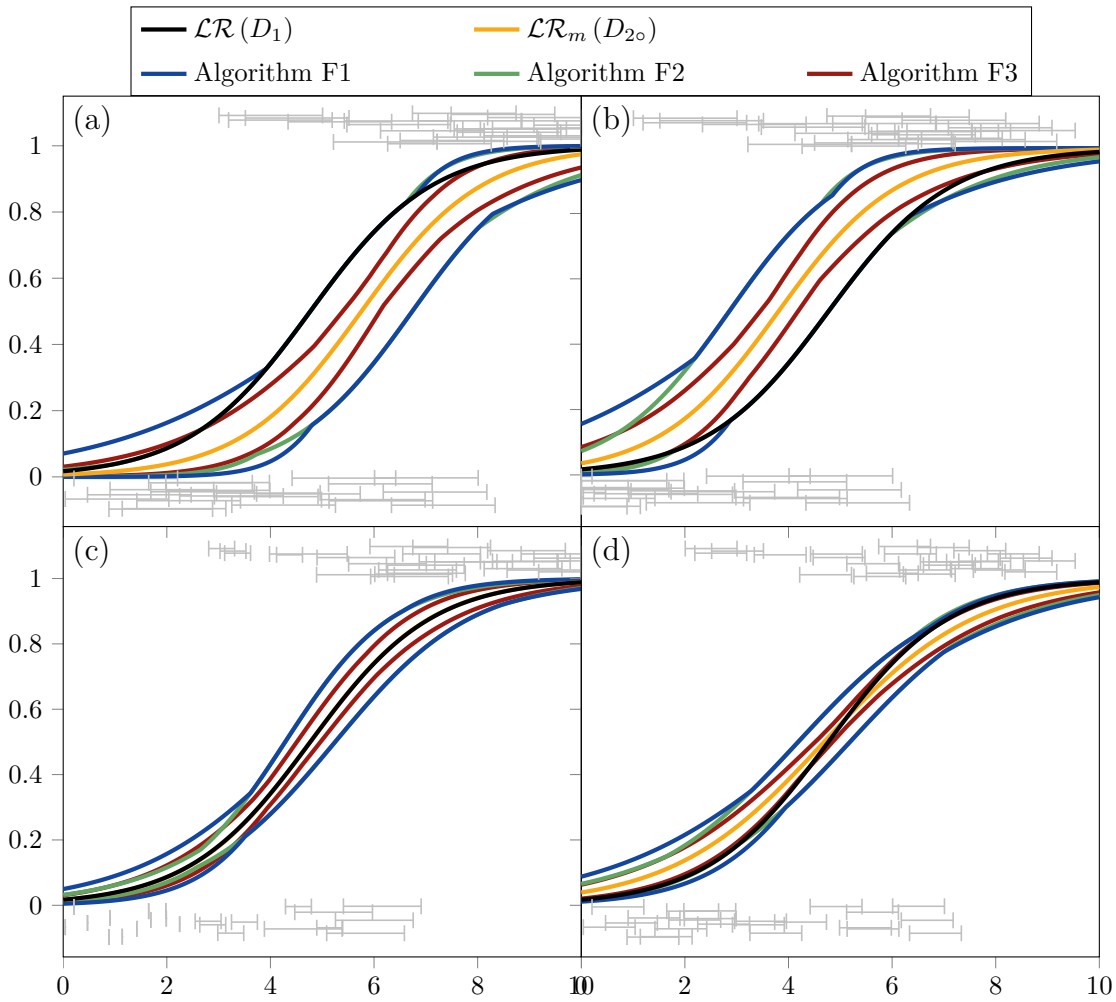

Figure 8: Logistic Regression models fitted on interval data that has been intervalised in the biased ways shown in Table 3. In all plots, $\mathcal{LR}(D_1)$ is the precise model shown in Figure 1.

| Plot | Intervalisation |
|------|-----------------|
| (a) | $D_{2a} = \{([x_i, x_i + 2], y_i) \ \forall (x_i, y_i) \in X\}$ |
| (b) | $D_{2b} = \{([x_i - 2, x_i], y_i) \ \forall (x_i, y_i) \in X\}$ |
| (c) | $D_{2c} = \left\{ \left( \begin{cases} x_i \text{ if } x_i < 2.5 \\ [m - 0.25, m + 0.25], \ m \in U(x_i \pm 0.25) \text{ if } 2.5 \leq x_i < 5.0 \\ [m - 0.75, m + 0.75], \ m \in U(x_i \pm 0.75) \text{ if } 5.0 \leq x_i < 7.5 \\ [x_i, x_i + 1.5], \text{ if } 7.5 < x_i \end{cases}, y_i \right) \forall (x_i, y_i) \in X \right\}$ |
| (d) | $D_{2d} = \left\{ \left( \begin{cases} [x_i, x_i + 1.5] \text{ if } y_i = 1 \\ [x_i - 1.5, x_i] \text{ otherwise} \end{cases}, y_i \right) \forall (x_i, y_i) \in X \right\}$ |

Table 3: Intervalisations used in Figure 8

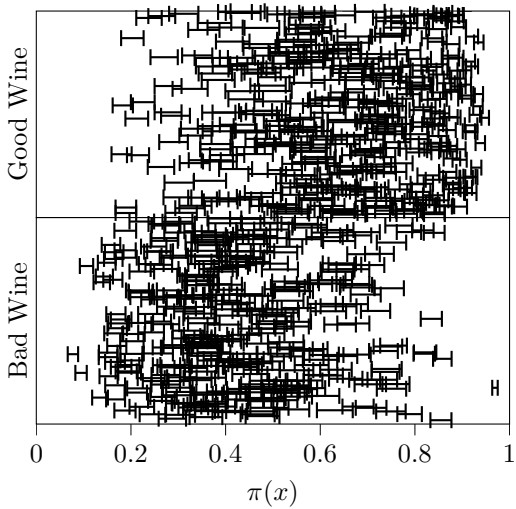
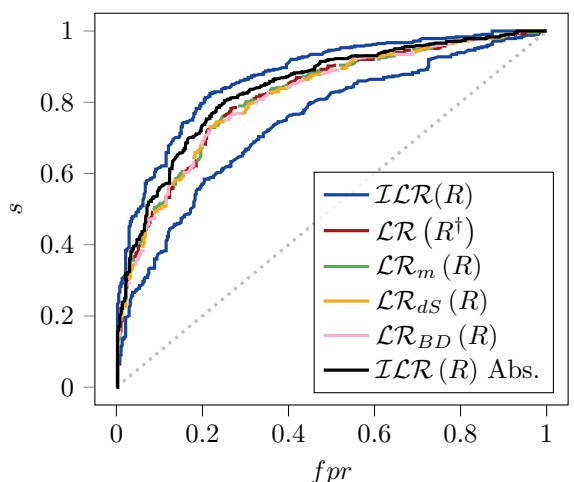

(a) Scatter plots of probability vs outcome for $\mathcal{ILR}\,(R)$

(b) Receiver operating characteristic curve for the simple example with added uncertain classifications.

Figure 9: Plots to show the discriminatory performance of the logistic regression models for the red wine example.

Classifications about whether a wine is good or not can be made. Selecting a threshold value of 0.5 gives the confusion matrices shown in Table 4. There are two possible interpretations of how the intervals can be expressed within the confusion matrices shown in a the intervals are plotted directly within the matrix giving the following statistics: $s = [0.652, 0.769]$ and $t = [0.707, 0.817]$. Allowing the model to abstain as in b implies that there are 91 wines for which a prediction could not be made solely as a result of the imprecision of the model. The summary statistics from this confusion matrix are: $s' = 0.738$, $t' = 0.795$, $\sigma = 0.117$ and $\tau = 0.110$.

|  | Good Wine | Bad Wine | Total |
|---|---|---|---|
| Predicted Good | [279,329] | [68,109] | [347,438] |
| Predicted Bad | [99,149] | [263,304] | [362,453] |
| Total | 428 | 372 | 800 |

(a) Tabulating inconclusive results as [0,1] intervals.

|  | Good Wine | Bad Wine | Total |
|---|---|---|---|
| Predicted Good | 279 | 68 | 347 |
| Predicted Bad | 99 | 263 | 362 |
| No Prediction | 50 | 41 | 91 |
| Total | 428 | 372 | 800 |

(b) Tabulating inconclusive results separately.

Table 4: Two possible confusion matrices for 100 test samples from the imprecise logistic regression model shown in Figure 3.

## 4    Uncertainty in Labels

This set-based approach can be extended to the situation where there is uncertainty about the outcome status meaning there are some points for which we do not know the binary classification and can be represented as the dunno interval $[0,1]$. In this situation the dataset $D$ contains $p$ variables with corresponding labels $(\mathbf{x}_1, y_1), (\mathbf{x}_2, y_2), \cdots, (\mathbf{x}_p, y_p)$ but also $q$ variables for which the label is unknown $(\mathbf{x}_{p+1}, \ ), (\mathbf{x}_{p+2}, \ ), \cdots, (\mathbf{x}_{p+q}, \ )$. For simplicity, we shall refer to the points with labels as being in set $d$ and those without labels in set $u$, $D = d \cup u$.

It is important to note that, whilst formally, all possible unobserved, and therefore unlabelled, $\mathbf{x}$ values could be considered to be in set $u$, however, this should not be considered the case. The datapoints that are in $u$ only contain $\mathbf{x}$ values that have been observed but for which the label is unidentified for some reason. This may be due to a participant dropping out of a medical trial before it concludes or there if there is disagreement between expert labellers about the true clasification. Thus it is not the case that one can increase the number of points in $u$ by introducing arbitrary new $\mathbf{x}$ values.

Traditional analysis may ignore all the points in $u$. However, they can be included within the analysis by considering the set of possible logistic regression models trained on all possible datasets that could be possible based upon the uncertainty. This set of datasets can be created by giving all unlabelled values the value 0, all unlabelled values the value 1 and all combinations thereof, i.e.

$$
\mathcal{ILR} = \left\{ \mathcal{LR}(D') \ \forall D' \in \left\{ \begin{array}{c} d \cup \{(\mathbf{x}_{p+1}, 0), \cdots, (\mathbf{x}_{p+q}, 0)\} \\ d \cup \{(\mathbf{x}_{p+1}, 0), \cdots, (\mathbf{x}_{p+q}, 1)\} \\ \vdots \\ d \cup \{(\mathbf{x}_{p+1}, 1), \cdots, (\mathbf{x}_{p+q}, 0)\} \\ d \cup \{(\mathbf{x}_{p+1}, 1), \cdots, (\mathbf{x}_{p+q}, 1)\} \end{array} \right\} \right\}. \tag{26}
$$

There are $2^q$ possible logistic regression models within this set. An imprecise logistic regression model can then be created by finding the envelope of the set, as shown in Algorithm L1. As the computational time for this algorithm increases as $\mathcal{O}(2^q)$, then as $q$ increases finding the bounds by calculating the envelope for all possible combinations can become computationally expensive.

Algorithm L2 reduces the number of iterations models that need to be fitted to find an *estimate* for the imprecise bounds. This algorithm first finds the logistic regression model assuming all uncertain labels are 0 and the logistic regression model assuming all uncertain labels are 1. The uncertain points are split into three groups: $G_1$ contains points which have a low $\pi$ value with both models, $G_2$ contains points which have a high $\pi$ value with both models and all other points in $G_3$. The algorithm assumes that the most extreme models can be found by giving all the points in $G_1$ the same label, all the points in $G_2$ the same label and only finding all possible combinations of labels for the points within $G_3$. This algorithm reduces the number of logistic regression models fitted to $2 + 2^{2+q'}$ where $q'$ is the number of points in $G_3$.

### 4.1 Alternative Methods

#### 4.1.1 Exclude Uncertain Results

The most straightforward approach to dealing with this is to remove the uncertain results from $D$ to produce $D_\times$ and a precise logistic regression model $\mathcal{LR}_\times(D)$. This approach may be valid if the missing data is small compared to the total dataset size and if it is missing at random or completely at random.

#### 4.1.2 Semi-Supervised Logistic Regression

Semi-supervised learning methods extend supervised learning techniques to cope with additional unlabelled data. Numerous authors present semi-supervised logistic regression methods based on a variety of different methods: Amini & Gallinari (2002) use Classification Expectation Maximisation; Krishnapuram et al. (2004) and Chi et al. (2019) use Bayesian methods; Bzdok et al. (2015) use an autoencoder and a factored logistic regression model; Chai et al. (2018) combine active learning and semi-supervised learning to " achieve better performances compared to the widely used semi-supervised learning and active learning methods." In all cases, it is important that the smoothness, clustering and manifold assumptions are valid to use semi-supervised learning techniques (Chapelle et al., 2006).

For this analysis, we have used the scikit-learns semi-supervised learning algorithm, which uses Yarowsky's algorithm to enable the logistic regression to learn from the unlabelled data (Yarowsky, 1995; Pedregosa et al., 2011).

---

**Algorithm L1:** Algorithm to find $\mathcal{ILR}(D)$ if $D$ has interval uncertainty within its labels.

---

**Input:** $d = \{(\mathbf{x}_i, y_i) \,\forall i = 0, \ldots, p\}$; $u = \{(\mathbf{x}_j, [0,1]) \,\forall j = 0, \ldots, q\}$; $D = d \cup u$

$\mathcal{ILR}(D) \leftarrow \{\}$;

**for all combinations** $C \in \{(0, \ldots, 0), (0, \ldots, 1), (1, \ldots, 0), (1, \ldots, 1), \cdots\}$ **do**

$\quad$ $D' \leftarrow d \cup \{(\mathbf{x}_j, C_j) \,\forall \mathbf{x}_j \in u\}$;

$\quad$ $\mathcal{ILR}(D) \leftarrow \mathcal{ILR}(D) \cup \mathcal{LR}(D')$;

**end**

**Output:** $\mathcal{ILR}(D)$

---

**Algorithm L2:** Algorithm to find $\mathcal{ILR}(D)$ if $D$ has interval uncertainty within its labels using heuristics to reduce the number of iterations needed.

---

**Input:** $d = \{(\mathbf{x}_i, y_i) \,\forall i = 0, \ldots, p\}$; $u = \{(\mathbf{x}_j, y_j = [0,1]) \,\forall j = 0, \ldots, q\}$; $D = d \cup u$

$D_{(1,\ldots,1)} \leftarrow d \cup \{(\mathbf{x}_j, 1) \,\forall \mathbf{x}_j \in u\}$;

$D_{(0,\ldots,0)} \leftarrow d \cup \{(\mathbf{x}_j, 0) \,\forall \mathbf{x}_j \in u\}$;

Find $\mathcal{LR}(D_{(1,\ldots,1)})$ and $\mathcal{LR}(D_{(0,\ldots,0)})$;

$G_1 \leftarrow \{\}$; $G_2 \leftarrow \{\}$; $G_3 \leftarrow \{\}$;

**for all** $u_j = (\mathbf{x}_j, y_j) \in u$ **do**

$\quad$ **if** $\pi_{\mathcal{LR}(D_{1,\ldots,1})}(\mathbf{x}_i) < 0.5$ *and* $\pi_{\mathcal{LR}(D_{0,\ldots,0})}(\mathbf{x}_i) < 0.5$ **then**

$\quad\quad$ $G_1 \leftarrow G_1 \cup \{u_j\}$

$\quad$ **else if** $\pi_{\mathcal{LR}(D_{1,\ldots,1})}(\mathbf{x}_i) > 0.5$ *and* $\pi_{\mathcal{LR}(D_{0,\ldots,0})}(\mathbf{x}_i) > 0.5$ **then**

$\quad\quad$ $G_2 \leftarrow G_2 \cup \{u_j\}$

$\quad$ **else**

$\quad\quad$ $G_3 \leftarrow G_3 \cup \{u_j\}$

$\quad$ **end**

**end**

**for all** $A$ *in* $\{0, 1\}$ **do**

$\quad$ **for all** $B$ *in* $\{0, 1\}$ **do**

$\quad\quad$ **for all combinations** $C \in \{(0, \ldots, 0), (0, \ldots, 1), (1, \ldots, 0), (1, \ldots, 1), \cdots\}$ **do**

$\quad\quad\quad$ $D' \leftarrow d \cup \{(\mathbf{x}_j, A) \,\forall (\mathbf{x}_j, y_j) \in G_1\} \cup \{(\mathbf{x}_j, B) \,\forall (\mathbf{x}_j, y_j) \in G_2\} \cup \{(\mathbf{x}_j, C_j) \,\forall (\mathbf{x}_j, y_j) \in G_3\}$;

$\quad\quad\quad$ $\mathcal{ILR}(D) \leftarrow \mathcal{ILR}(D) \cup \mathcal{LR}(D')$;

$\quad\quad$ **end**

$\quad$ **end**

**end**

**Output:** $\mathcal{ILR}(D)$

---

## 4.2   Demonstration

Dataset $D_3$ has been created from dataset $D_1$ by replacing five labels from the dataset with the $[0,1]$ interval. The labels that have been changed are around the point at which the data goes from 0 to 1. This dataset is shown in Figure 10, with the uncertain labels plotted as vertical lines. The figure shows all the logistic regression models that have been fitted using both Algorithms L1 (grey lines) and L2 (coloured). $\mathcal{ILR}(D_3)$ is the envelope of these sets and is shown with the black dashed lines. Since the black lines always correspond to the extremum of the colour lines, Algorithm L2 has correctly estimated the imprecise bounds, and any interval $\pi$ value found from imprecise models is guaranteed to contain the true value.

Figure 11 shows the imprecise logistic regression model that is trained on this uncertain dataset, and, for comparison, the model trained on the dataset with the dunno labels removed, $\mathcal{LR}_\times(D_3)$, and the semi-supervised model, $\mathcal{LR}_{ss}(D_3)$. From the figure, it is also notable that $\mathcal{LR}_\times(D_3)$ and $\mathcal{LR}_{ss}(D_3)$ are similar.

As in Section 3.3, it is helpful to consider different scenarios within which the labels have been removed. Figure 12 shows four different scenarios within which the data has been made uncertain, and the imprecise logistic regression models have been fitted using Algorithm L2. Using this algorithm allowed plot (b) to be computed since Algorithm L1 would have required $2^{20}$ models to be fitted and have been computationally prohibitive. In all four plots, the imprecise model bounds the base model. It is also notable that the semi-supervised and discarded data approaches are similar in all the plots.

This plot demonstrates that if it can be assumed that the labels are missing at random, as in (a) and (b), the two alternative approaches are reasonably close to the true model. However, if this is not the case, then significant differences between the approaches and the imprecise method must be used to obtain a model that is guaranteed to bound the true model.

## 4.3   White Wine Example

As in Section 3.4 it is useful to consider this methodology on a real dataset. In this instance, we will use the white wine dataset from Cortez et al. (2009). This dataset contains the same covariates as the red wine dataset used in Section 3.4 but contains many more samples (4898). Again good wine has been defined as having a quality $\geq 6$. This data has been split into training and test samples, with 1618 and 3281 samples, respectively. In order to simulate sommeliers being unsure about the classification of marginally good wine, 100 samples with quality=6 have had their labels removed. Let $W$ be the uncertain dataset. Algorithm L2 can be used to fit the imprecise logistic regression model on this dataset. For comparison, $\mathcal{LR}_{ss}(W)$ and $\mathcal{LR}_\times(W)$ have also be found.

The discrimination plots for these models are shown in Figure 13. Figure 13a shows that very few points have been given a low probability of being bad wine. Most of the bad wine has $\pi \approx 0.5$ whereas good wine has a high probability ($\pi \approx 0.9$). This plot suggests that when making classifications from the model, selecting a threshold value of $C = 0.7$ would be an appropriate choice to distinguish between the two classes. ROC curves can also be plotted. As with the previous examples, the precise models all have very similar curves and AUC values which the abstaining model 'beats'. (In this case $AUC[\mathcal{ILR}(W)] = [0.716, 0.826]$, $AUC[\mathcal{ILR}(W)(\,Abstain)] = 0.794$, $AUC[\mathcal{LR}_{ss}(W)] \approx AUC[\mathcal{LR}_\times(W)] \approx AUC[\mathcal{LR}(W^\dagger)] = 0.774$)

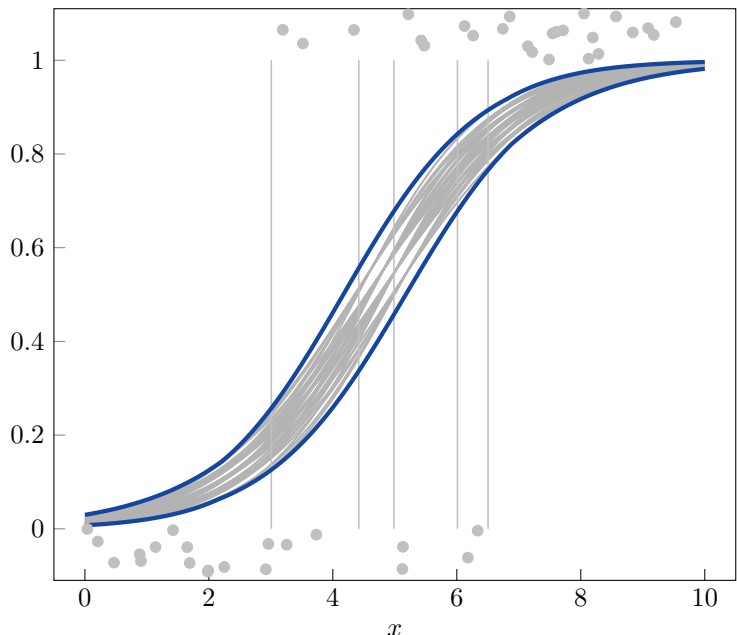

Figure 10: Imprecise logistic regression model with the uncertain labels represented by the vertical lines. The grey lines represent all possible models found using Algorithm L1. The blue lines are the bounds found using Algorithm L2.

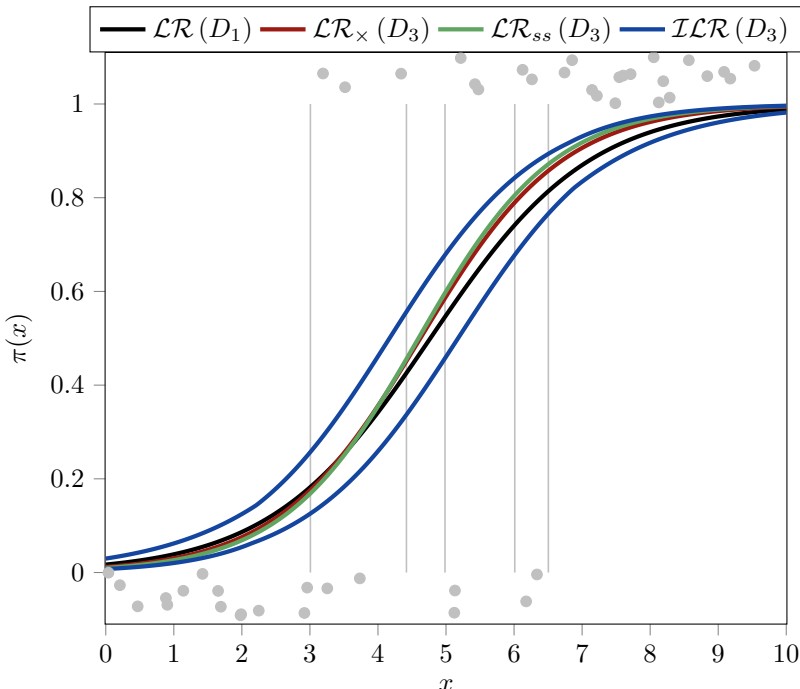

Figure 11: Bounds for the imprecise logistic regression or all 50 points with 5 points made uncertain. Compared with $\mathcal{LR}_\times (D_3)$ and $\mathcal{LR}_{ss} (D_3)$.

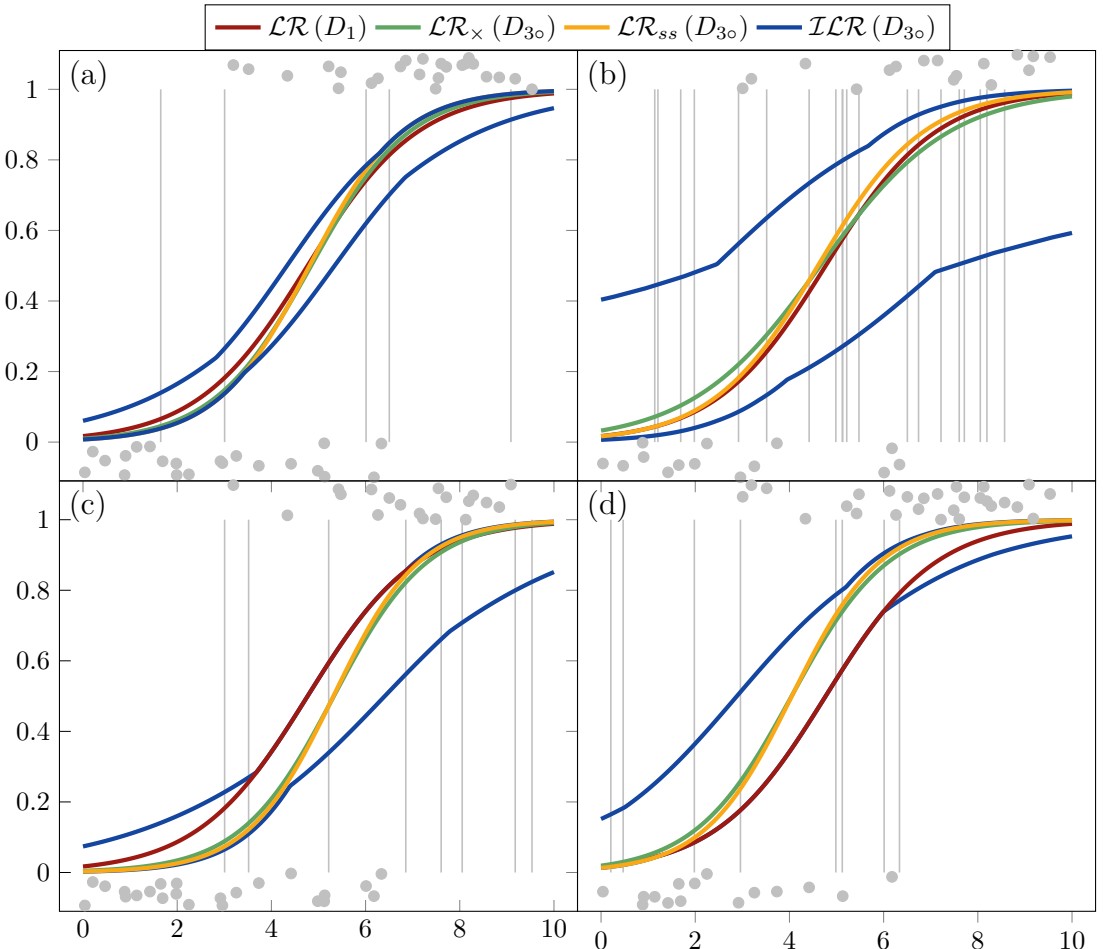

Figure 12: 4 different scenarios within which dataset $D_1$ has had labels removed. $D_{3a}$ has 5 labels missing at random. $D_{3b}$ has 10 labels missing at random. $D_{3c}$ has 8 1 labels missing. $D_{3d}$ has 8 0 labels missing.

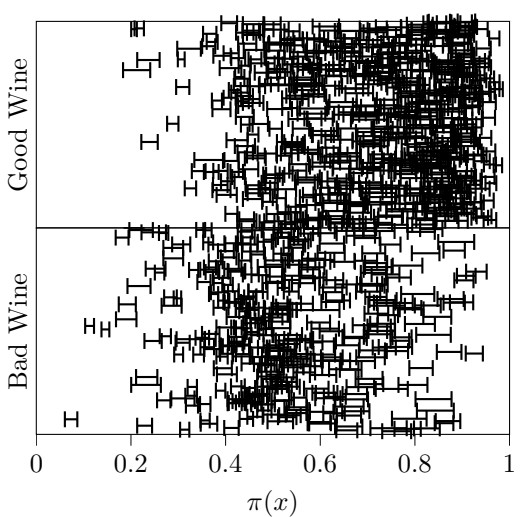

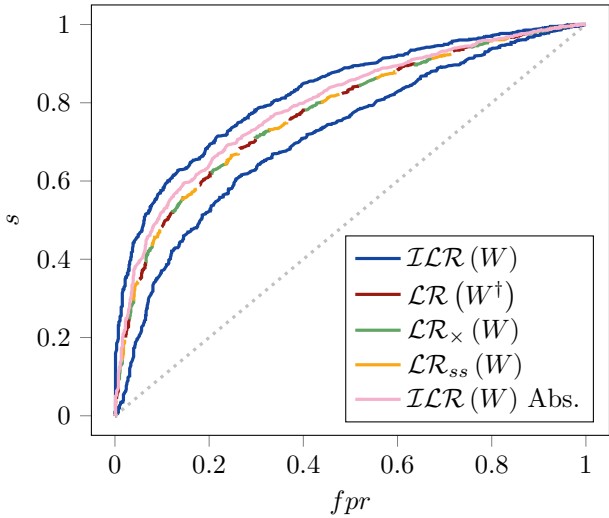

(a) Scatter plots of probability vs outcome for $\mathcal{ILR}\,(W)$

(b) Receiver operating characteristic curve for the simple example with added uncertain classifications.

Figure 13: Plots to show the discriminatory performance of the logistic regression models for the white wine example.

# 5 Uncertainty in Both Features and Labels

The imprecise approach can be used when there is uncertainty about both the features and the labels. Such situations are present in numerous real-world datasets. An imprecise logistic regression model can be found in this scenario through a combination of the algorithms in Sections F1 and 4.

## 5.1 Example

Osler et al. (2010) use a logistic regression model to predict the probability of death for a patient after a burn injury. The model they use is based upon a subset of data from the American Burn Association's National Burn Database[2]. The dataset has a mix of discrete (gender, race, flame involved in injury, inhalation injury) and continuous variables (age, percentage burn surface area) that can be used to model the probability that a person dies (outcome 1) after suffering a burn injury. Osler et al. exclude some patients from the dataset before training their model. They remove patients if their age or 'presence of inhalation injury' was not recorded. Additionally, as patients older than 89 were assigned to a single age category in the original dataset, they gave them a random age between 90 and 100 years.

Osler et al. did not need to exclude these patients merely because of epistemic uncertainty about the values. The proposed approach can be used with the original data. For instance, patients for which the outcome was unknown could have been included within their analysis as described in Section 4. Similarly, patients whose inhalation injury or age was unknown could have been included with the method described in Section 3. Patients with unknown inhalation injury could have been included as the $[0, 1]$ interval. Patients whose age was completely unknown could have been replaced by an interval between the minimum and maximum age, whereas if there was uncertainty because they were over 90 years old, then they could be intervalised as $[90, 100]$.

Other interval uncertainties may be present within the dataset. It is unlikely to be the case that all the people used within the study fit neatly into the discrete variables given. For instance the variable race is valued at 0 for "non-whites" and 1 for "whites". However, it goes without saying that the diversity of humanity does not simply fall into such overly simplified categories; there are likely to be many people who could not be given a value of 0 or 1 and should instead have a $[0, 1]$ value. The same is true for gender. Not everyone can be defined as male or female. Also, there is almost certainly some measurement uncertainty associated with calculating the burn surface area that may also be best expressed as intervals. For simplicity, these uncertainties have not been addressed below.

For this analysis, the subsample of the dataset used by Osler et al. made available by Hosmer Jr et al. (2013, p. 27) has been used. This version of the dataset includes 1000 patients from the 40,000 within the entire study and has a much higher prevalence of death than the original dataset. Because access to the original data is prohibitively expensive, the values in this dataset have been re-intervalised to replicate some of the removed uncertainty to create a hypothetical dataset, $B$, for this exposition. As there are no individuals older than 90 within the dataset, that particular re-intervalisation has not been possible, so all patients older than 80 have had their ages intervalised as [80,90]. Similarly, for 20 patients, the censored inhalation injury has been restored to dunno interval. Ten patients, who had been dropped because their outcome status was unknown, have been restored with status represented as [0,1].

There are two possible routes for an analyst to proceed when faced with such a dataset. They could follow the original methodology of Osler et al. and randomly assign patients with interval ages a precise value and then discard all other patients for which there is some uncertainty. Alternatively, the analyst could include the uncertainty within the model by creating an imprecise logistic regression model. As there is uncertainty within both the features and the labels, the model can be estimated by finding the values within the intervals that correspond to the minimum and maximum for $\beta_0$, $\beta_1$, etc. $\mathcal{ILR}(B)$ is the imprecise logistic regression fitted from this burn data. For comparison, $\mathcal{LR}(B_\times)$ has also been fitted based on removing the uncertainty in $B$ using the same methodology as Osler.

---

[2] http://ameriburn.org/research/burn-dataset/

Regarding the performance of the two models, we can again turn to visualisations, as shown in Figure 14. Firstly, looking at Figure 14c we can see that the vast majority of patients who were given a low probability of death ($\pi$) did indeed survive, and patients who were given a high probability of death did sadly die. The ROC plots are shown in Figure 14a, Figure 14b shows the upper right corner of the plot in more detail. $\mathcal{ILR}(B)$ has a $AUC = [0.955, 0.974]$, the no prediction model has $AUC = 0.972$ and $\mathcal{LR}(B)$ has $AUC = 0.966$.

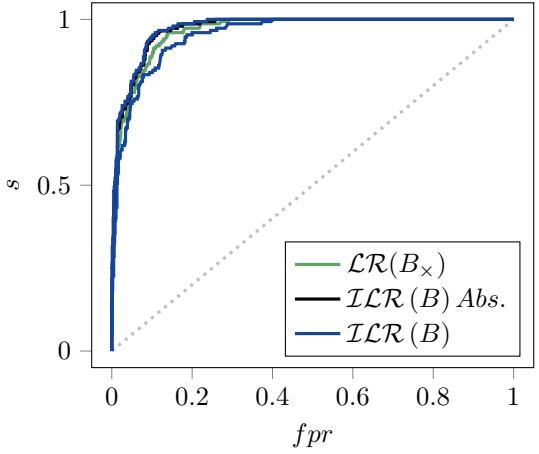

(a) Receiver operating characteristic curves for the burn example.

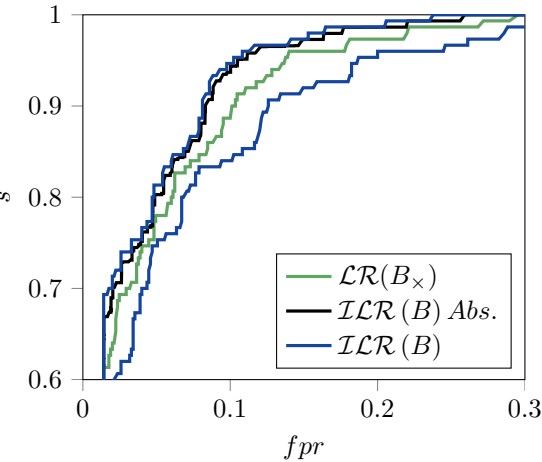

(b) Receiver operating characteristic curves for the burn example.

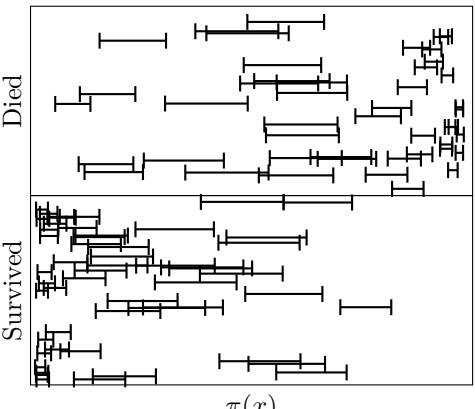

(c) Scatter plots of probability vs outcome. The two outcomes have been separated into different plots for clarity.

Figure 14: Plots to show the discriminatory performance of the various logistic regression models for the burn survivability example

It is pertinent to consider how a model is likely to be used and how uncertainty about the predicted probability of death impacts the classification. One method of dealing with this uncertainty that arises in Sections 4 and 3 is not making a prediction when the interval for $\pi$ straddles $C$. This method may not be appropriate in this example. What should happen when the model is unable to make a prediction should depend on what the result of deciding a patient has a high risk of death means clinically. If the model was being used to triage patients that need to go to a major trauma centre because the probability of death is considered high, then–out of an abundance of caution–one might prefer that if any part of the interval probability was greater than some threshold, the patient should be considered high risk. Equivalent to taking the probabilities from

the upper bound of the range,

$$\text{high risk} = \begin{cases} 1, & \text{if } \overline{\pi} \geq C \\ 0, & \text{otherwise.} \end{cases} \tag{27}$$

However, if patients who are considered high risk then undergo some life-altering treatment that is perhaps only preferable to death, then under the foundational medical aphorism of "first do no harm", it may be preferable to consider a patient high-risk only if the whole interval is greater than the decision threshold, this is equivalent of taking the probabilities from the lower bound of the range,

$$\text{high risk} = \begin{cases} 1, & \text{if } \underline{\pi} \geq C \\ 0, & \text{otherwise.} \end{cases} \tag{28}$$

Using the imprecise model in these scenarios would lead to better outcomes as the epistemic uncertainty would not be ignored. It is also the case that a patient that has a wide interval (as is the case for some in Figure 14c), implying that there is large epistemic uncertainty about the prediction, the medics would be aware of the uncertainty associated with the model and therefore may prefer to decide another way.

## 6 Discussion and Conclusion

Analysts face uncertainties in the measurement values of their data. Often this uncertainty is either assumed away or just completely ignored. However, it may be better to compute with what we know rather than to make assumptions that may need to be revised later. Many uncertainties are naturally expressed as intervals arising from measurement errors and missing or censored values. In the case of logistic regression, when faced with interval uncertainties, samples are often dropped from analyses—assuming that they are missing at random—or reduced down to a single value. Analysts should not simply throw this uncertainty away to make subsequent calculations easier.

Interval uncertainties can be included within logistic regression models by considering the set of possible models as an imprecise structure, including in situations where there is uncertainty about the binary outcome status. The present analysis showed that it is not reasonable to throw away data when the status is unknown if the reason the data has gone missing is dependent on the value or status of the missing samples. The case studies showed that previous methods used to handle interval uncertainties are ill-suited to situations where the narrow assumptions that they rely upon are unjustified or untenable. The methods based upon imprecise probabilities described in this paper work whenever there are interval uncertainties in the data, regardless of how they happened to arise.

The imprecise approach presented within this paper introduces two distinct uncertainties within logistic regression models. The first is the uncertainty about what the expected label for a particular $\mathbf{x}$ should be, expressed in an aleatoric way by considering $\pi(\mathbf{x})$ as a probability. The second type of uncertainty, added by imprecise logistic regression models, is the epistemic uncertainty expressed by the interval $\pi(\mathbf{x})$ values.

A reviewer of this paper noted that, in the case of label uncertainty, it seems counterintuitive that adding unlabelled examples yields an important uncertainty, despite the fact that the unlabelled examples appear uninformative and thus useless. However, it is essential to consider that there is a critical distinction between not knowing a label for some arbitrary value and knowing that we do not know a label for a directly observed value. In the latter case, there is information in the fact that it is known that the label is unknown. As shown within Figure 12, if an analyst can not assume that the labels are missing (completely) at random, they cannot say that they are uninformative.

Additionally, it is not the case that these uncertain points are being 'added'. They are not being removed or assumed away, as would have been the case in traditional analyses. Previously, analysts did not have methods to characterise this epistemic uncertainty, whereas the presented methods enable analysts to keep their interval data and not neglect them.

The situation may occur when one dataset contains precisely known values and another dataset with interval data. Whilst it may seem obvious to pool this data into a big dataset, it may not be the case that bigger

data is better data, especially if there is more uncertainty in the pooled dataset. For a full discussion about combining datasets with different levels of imprecision, see Tretiak & Ferson (2022).

If the uncertainty within the dataset can be assumed to be missing (completely) at random, the intervals are small compared to the underlying size of the data, or the equidistribution hypothesis holds. Analysts may be justified in not performing an imprecise logistic regression and using one of the alternative methods reviewed within this paper. However, if there is doubt about whether these assumptions hold, then the methods presented within this paper must be used to characterise this uncertainty. It has always been easy to get wrong answers; thus, whilst the algorithms presented within this paper are computationally expensive, only they will account for the full uncertainty within the dataset.

When using the new approach to classify, each new sample gets an interval probability of belonging to one of the binary classifications. Therefore, there are likely to be samples for which a definitive prediction cannot be made. If an analyst is happy to accept a *don't know* result, then the regression's performance as a classifier may be improved for the samples for which a prediction is made. It may seem counterproductive or unhelpful for a model to return a don't know result. However, this can be desirable behaviour; saying "I don't know" is perfectly valid in situations where the uncertainty is large enough that a different decision could have been reached. Uncertainty in the output can allow for decisions made by algorithms to be more humane by requiring further interrogation to make a classification. Alternatively, depending on the use case, other ways of making decisions based on uncertain predictions could be made, such as being conservative or cost-minimising. Although deciding indeterminate predictions at random would be capricious.

Within many active learning systems, the model is already forced to abstain from predicting labels for samples when the probability is close to the decision boundary ($\pi(x) \approx C$) so that a human can provide a classification (Schein & Ungar, 2007; Chai et al., 2018). If the imprecise logistic regression model presented in this paper were used in such a system, it would have the advantage of clearly providing a criteria for which abstentions are prefered, as opposed to a post-hoc decision based upon an arbitrary definition of how close to the boundary is too close. Additionally, this method exposes samples for which the model returns broad interval probabilities even if their centre is not close to the decision boundary and would have been considered a clear decision previously. If an abstention region is provided, then any interval probability produced from the model that straddles the region would be indeterminate.

We have shown that it is possible to include interval uncertainty in both outcome status and predictor variables within logistic regression analysis by considering the set of possible models as an imprecise structure. Such a method clearly can express the epistemic uncertainties within the dataset that are removed by traditional methods. Future work should be invested in finding improved algorithms to make them less computationally expensive for large-scale datasets.

## A   Illustration that Equation 19 approximates the envelope of Equation 7

To illustrate, it is useful to first consider a linear regression model with intervals in the dependent variables. Consider the two interval datapoints $(x_1, y_1) = ([1, 2], 2)$ and $(x_2, y_2) = ([3, 4], 3)$

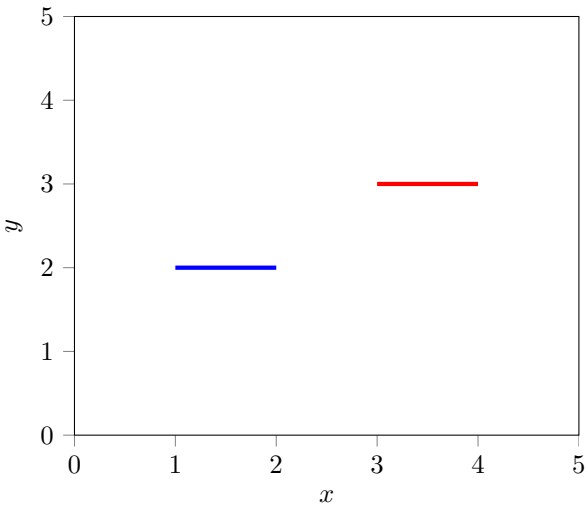

Figure A1: Intervals points $(x_1, y_1) = ([1, 2], 2)$ and $(x_2, y_2) = ([3, 4], 3)$

Whilst there are infinitely many linear regression models $(y = \beta_0 + \beta_1 x)$ that could be drawn by selecting points from $x_1$ and $x_2$ there are four extreme values as shown in A2. In this instance, the four regression lines are:

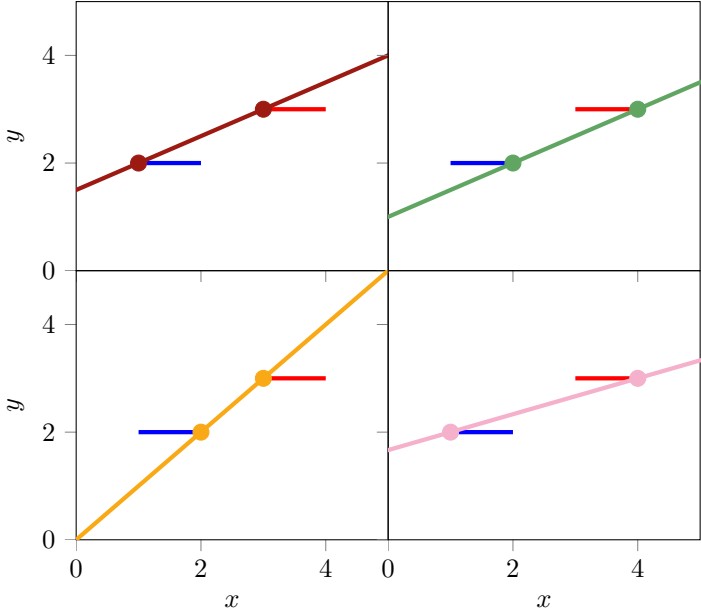

Figure A2: 4 extreme regression models that can be fitted from the two intervals

$$y = x \tag{29a}$$

$$y = \frac{1}{3}x + \frac{5}{3} \tag{29b}$$

$$y = x - 2 \tag{29c}$$

$$y = \frac{1}{2}x + \frac{3}{2} \tag{29d}$$

If we consider these four lines as a set and consider the imprecise regression model as the envelope of this set, then we get the band shown in Figure A3. Six segments make up this band ($AB$, $BC$, $CD$, $EF$, $FG$, $GH$). These lines correspond to lines with the minimum and maximum $\beta_0$ and $\beta_1$ values plus the line drawn with the left-most and right-most values from within the intervals.

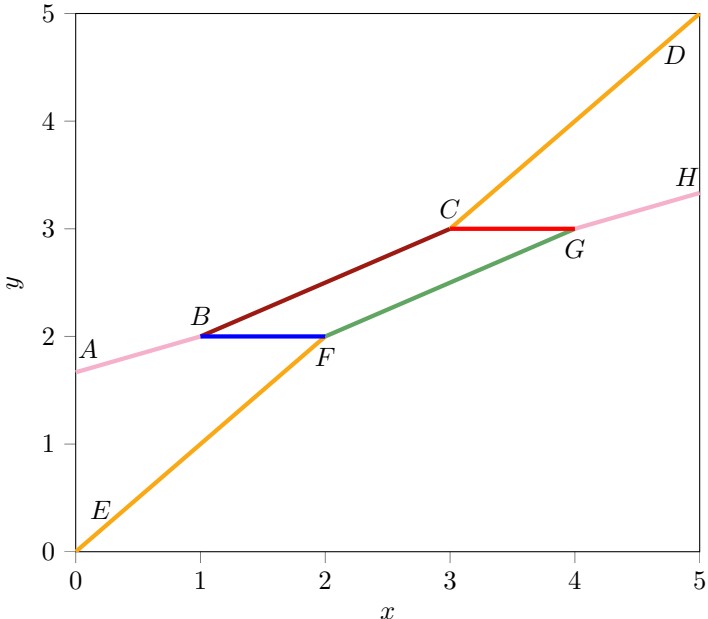

Figure A3: Envolope of the extreme lines shown in Figure A2

In theory, there are six possible models; it just so happens to be the case that the line which contains $\underline{\beta_0}$ also has $\overline{\beta_1}$ and vice versa. Figure A4 shows 100 regression models fitted using 100 Monte Carlo samples for values within the intervals. The whole band between the black lines would have been filled with enough samples or systematic sampling.

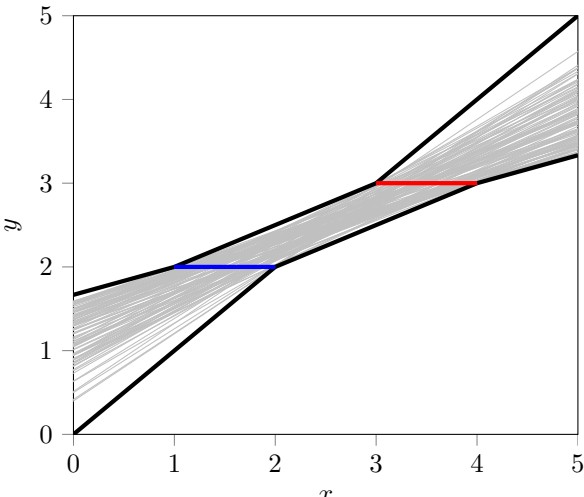

Figure A4: 100 regression models fitted on Monte Carlo samples for values within the intervals.

If there are four instead of two intervals, we can test how well the method finds the minimum and maximum coefficient values alongside the all-left bound, and all-right bound models. Figure A5 shows four intervals and the band bounded by the six lines suggested to be plotted. As before, we can sample values from within the intervals to check the performance of the band.

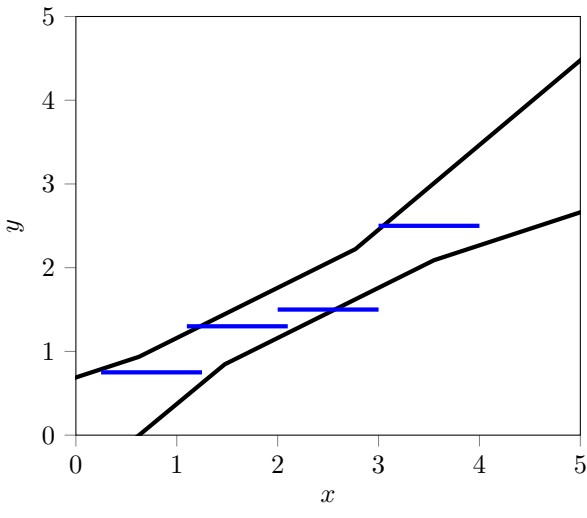

Figure A5

Again we can test whether this imprecise model bounds all possible regression models for values within the intervals. If we do a systematic sample of points within the four intervals, we can find all valid linear regression models consistent with the intervals. The envelope of these models is shown with the grey band in Figure A6. As can be seen, this band is not entirely within the black lines. For the model shown in Figure A6, doing this gives a value $A = 0.0311$.

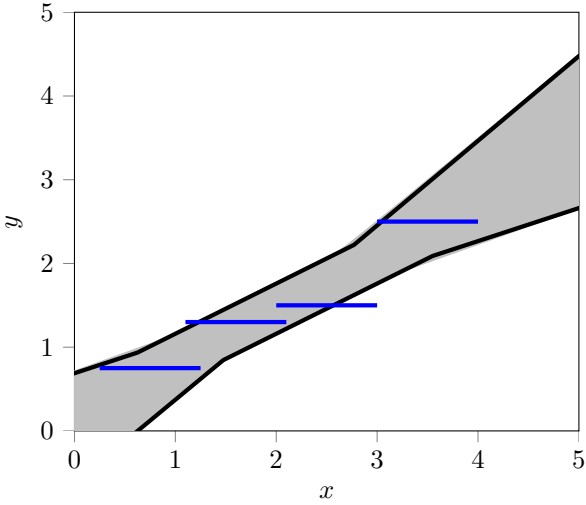

Figure A6

For logistic regression, since Equation 1 can be written as

$$\pi(x) = \frac{1}{1 + \exp{-r(x)}} = \sigma(r(x)) \tag{30}$$

where $r(x) = \beta_0 + \beta_1 x$ and $\sigma$ is the sigmoid function. Since we know that the 6-line method is suitable for estimating the envelope of $r(x)$ and the sigmoid function is monotonic, we can use this method for Logistic Regression. The represents the set represented by Equation 7 produced by Algorithm F1.

If we consider two intervals with opposite binary labels $(x_1, y_1) = ([1, 2], 0)$ and $(x_2, y_2) = ([3, 4], 1)$, shown in Figure A7. Again we can consider that the endpoints of the intervals will produce the minimum and

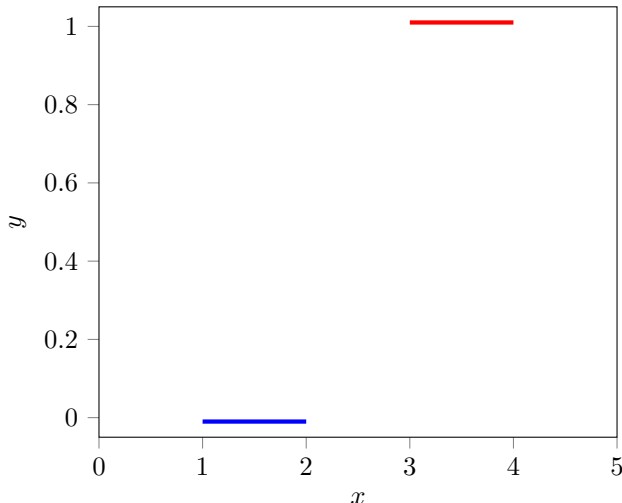

Figure A7: Two intervals with binary labels

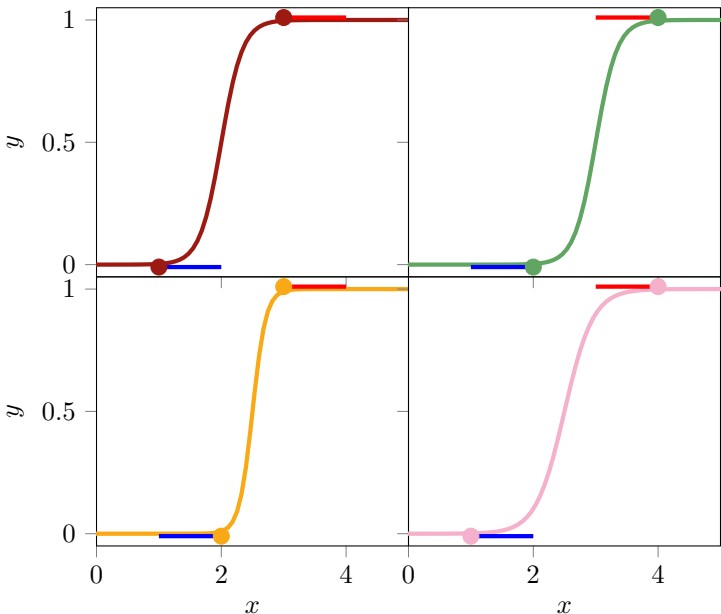

Figure A8: 4 extreme regression models that can be fitted from the two intervals.

maximum logistic regression models, as shown in Figure A8. As before, these lines represent the minimum and maximum coefficient values possible.

Again we can test whether the envelope of these four models (in this case, just the two lines that represent the models fitted on the endpoints of the intervals) will contain all possible models for values within the intervals via systematic sampling. This is shown in Figure A9. If we move to an imprecise logistic regression model fitted on four intervals using Algorithm F1, as shown in Figure A10. We can again test using a systematic sampling of the intervals whether the model fully covers all possible models (shown within the grey bounds). As before, the imprecise model does not perfectly model the systematic plot. In this instance, we have a value of $A = 0.0252$.

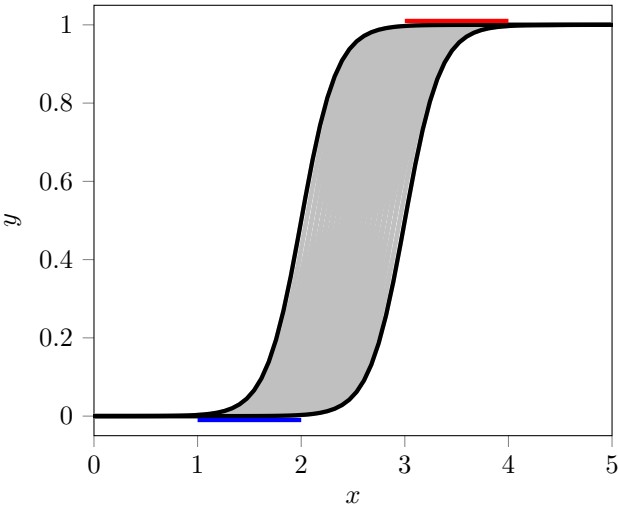

Figure A9: Envelope of the logistic regression models shown in Figure A8 and models produced by systematically sampling the intervals.

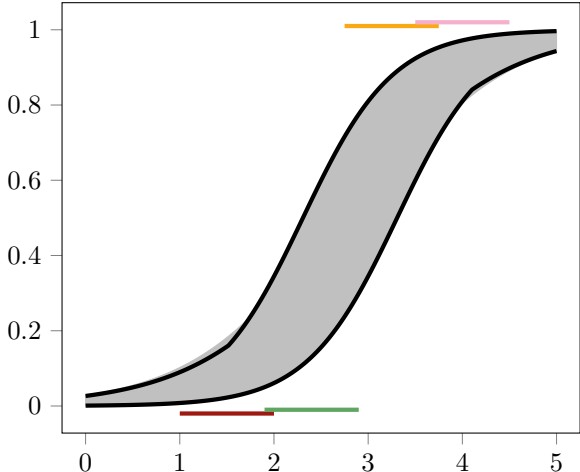

Figure A10: Imprecise Logistic Regression model for the four intervals

## B    Illustration that Algorithm F2 approximates the envelope of Equation 7

If we reconsider A8 but instead of considering that the four lines correspond to the minimum and maximum intercept and coefficient values, note that, as shown in Figure B1, the lines produced also represent the minimum and maximum spread of values around $x = 1$, $x = 2.5$ and $x = 4$.

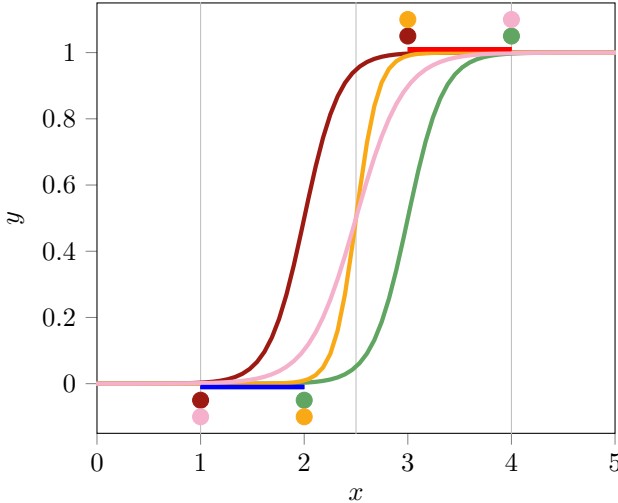

Figure B1: Reproduction of Figure A8 showing that the extreme models can be considered as the lines that correspond to the minimum and maximum spread of values around $x = 1$, $x = 2.5$ and $x = 4$ (shown with grey lines).

This leads to another way of approximating Equation 7. If we $x = 1$ and $x = 4$ are the minimum and maximum values from within the dataset, the minimum and maximum spread around these points represent the all-left and all-right bound models. The $x = 2.5$ corresponds to the minimum and maximum slope of the logistic regression curve. Finding the $x$ value that corresponds to this extreme case is trivial when there are only two intervals however is challenging for more complicated datasets.

One approach is to select $P$ values between the minimum and maximum values from the dataset and the minimum and maximum values themselves, giving $2 + P$ values to the sample. For a dataset with $m$ features, this leads to $(2 + P)^m$ models that would need to be found.

An alternative approach is first to fit the all-left and all-right models ($\mathcal{LR}\left(\underline{D}\right)$ and $\mathcal{LR}\left(\overline{D}\right)$), then select is to select $P$ $\pi$ values ($\left\{\frac{p}{P+1} \; \forall p = 1, \ldots, P\right\}$). For each of these $P$ values, we can find $\underline{T}$ such that $\pi_{\mathcal{LR}(\underline{D})}(T) = P$ and $\overline{T}$ such that $\pi_{\mathcal{LR}(\overline{D})}(T) = P$ we can then find the datasets that correspond to the minimum and maximum spread of values using Algorithm 1 and 2 respectively. This approach has complexity $2(1 + P)$ and is the approach described within Algorithm F2.

---

**Algorithm 1:** Procedure to find the minimum spread of the intervals in $D$ around $T$.

---

**Data:** $D, T$
**for all Intervals** $(I = \left[\underline{i}, \overline{i}\right])$ **in $D$ do**
  **if** $T \in P$ **then**
  |    $D_{min}(I) = T$;
  **else if** $T < P$ **then**
  |    $D_{min}(I) = \underline{i}$;
  **else**
  |    $D_{min}(I) = \overline{i}$;
**Output:** $D_{min}$

---

---

**Algorithm 2:** Procedure to find the maximum spread of the intervals in $D$ around $T$.

---

**Data:** $D$, $T$
**for all Intervals** $(I = \left[\underline{i}, \overline{i}\right])$ **in** $D$ **do**
  **if** $T \in P$ **then**
    **if** $|\underline{i} - T| > |\overline{i} - T|$ **then**
      $D_{max}(I) = \underline{i}$;
    **else**
      $D_{max}(I) = \overline{i}$
  **else if** $T < P$ **then**
    $D_{max}(I) = \overline{i}$;
  **else**
    $D_{max}(I) = \underline{i}$;

**Output:** $D_{max}$

---

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
