# OpenReview forum: "Logistic Regression Through the Veil of Imprecise Data"
_TMLR — Rejected by TMLR_

### Review · Reviewer_K6x7 · 2022-10-11

**Summary Of Contributions:**

The paper proposes novel methods to quantify the predictive uncertainty of logistic regression in scenarios where the training data have uncertainty about feature values, labels, or both. Feature uncertainty is assumed to be represented as intervals that must contain the true value of the feature. Label uncertainty is considered through some data being unlabelled, i.e. using the semi-supervised setting. The methods are demonstrated on various small datasets, showing the resulting predictive intervals visually as prediction bands.


**Broader Impact Concerns:**

I do not have any concerns on the ethical implications of this work.


**Requested Changes:**

In my opinion, all the weaknesses highlighted above are critical and need addressing before acceptance can be considered. As a result, it must become clear (or at least clearer) to what extent the intervals provided by the models are valid and how much
better the proposed methods are compared to existing methods and simpler baselines.

**Strengths And Weaknesses:**

Strengths:
* The paper provides clear motivation and highlights the shortcomings in existing methods;
* The existing literature and methods are covered in sufficient detail;
* The methods are demonstrated on several datasets visually, helping readers to develop intuition about the matter.

Weaknesses:
* In the task of finding an interval $\pi(x)$ of minimum and maximum predictions across the set of all models within ILR(D) as defined in Eq.(7), the authors claim that it is only necessary to consider a small number of models as defined in Eq.(8). However, no proof is given for this claim, and it has not been checked empirically either whether this is true. This is a critical shortcoming because it leaves it unclear whether the bounds obtained by Eq.(8) or Algorithm 1 are valid or not. In other words, is it possible to have a model which belongs to ILR(D) as defined by Eq.(7) but which provides predictions that are occasionally out of the bounds determined by ILR(D) of Eq.(8) or by ILR(D) of Algorithm 1? Figure 3 shows an example where 60 models randomly drawn from the interval (i.e. from the set of Eq.(7)) are shown all to fall between the extremes defined by Algorithm 1 (which approximates Eq.(8)). However, 60 is a small number; did the authors check with thousands of models as well? Can it be proved that all models of Eq.(7) fall between the extremes of Eq.(8)? If not, then have the authors been able to find a counterexample? Figure 3 is about a dataset with a single feature, but are the models of Eq.(7) within the bounds defined by Eq.(8) when there are more features? Can this be proved? If not, can the authors demonstrate a counterexample? The methods of this paper can still be valuable if the bounds are not entirely valid; in this case, they are just a heuristic approximation and might still be useful
in practice. However, this has to be then validated by extensive experiments, exploring how strongly and how often the bounds are violated. Otherwise, it would be very hard to trust a model for which there is no understanding of its capabilities and limitations. In such experiments, it would also be important to compare to simpler baseline methods, such as randomly drawing a fixed number of models from within ILR(D) of Eq.(7) and using the set of models trained on these datasets (essentially the same as the 60 models in Figure 3). Would the method of Algorithm 1 provide typically wider intervals, given a comparable amount of computational resources? As a slightly more advanced baseline, what if the models are drawn from within ILR(D) of Eq.(7) not entirely randomly but such that each feature has the value of one or the other extreme (randomly chosen which one)?

* Related to the previous shortcoming, the paper currently makes unjustified claims as if there were a guarantee, e.g. "only the imprecise method guarantees coverage of the true model" at the bottom of page 9. It only becomes a guarantee if it has been proved. Even if the feature coefficients beta of the true model fall between the minimal and maximal coefficients learned by stochastic optimisation, it is not obvious that the predictions also always fall into the bounds, because the combined effect of multiple betas has not been considered in Eq.(8).

* The paper has not quantified how expensive the computations in the proposed methods are in practice. This is fundamentally important because the goal of the proposed method is to save computation by considering fewer models of Eq.(8) instead of all models of Eq.(7). Instead of stochastic optimisation to find extreme values of beta coefficients, perhaps it is better to consider a larger random sample of models from Eq.(7)? Maybe not, but this needs demonstrating.

---

### Review · Reviewer_L5Uz · 2022-10-14

**Summary Of Contributions:**

The submission proposes algorithms for two-class logistic regression on interval-valued data. It also considers how to adapt evaluation measures such as precision and recall when intervals are generated, rather than point estimates of class probabilities.

**Requested Changes:**

Please address all the comments above as well as the (mainly) smaller issues listed below.

===

Abstract

First sentence of abstract: what about polytomous logistic regression?

"many datasets" - this seems hard to believe, it would be useful to give at least one example here already

Introduction

First sentence: what about polytomous logistic regression?

"where the time of the event is not essential." - delete?

"linearly related or equal variance" - grammar

"This assumption is valid when the sampling uncertainty or natural variability in the
data is significant compared to the epistemic uncertainty or if values are missing at random (Ferson et al.,
2007)." - it is not clear to me why this is the case and Ferson et al. is just a tech report; please explain in the paper

"Measurement uncertainty is sometimes best represented
as intervals, sometimes called "censored data"." - this is not what censoring normally refers to

"to Simplify"

"While these approaches are computationally expedient, they underrepresent the imprecision by presenting a
single middle-of-the-road logistic regression." - please explain why this applies to the approach based on the equidistribution hypothesis

"The use ... has been used ..."

"The imprecise probabilities approach makes the fewest assumptions, but some statistics can be
computationally challenging for large datasets (Ferson et al., 2007)." - more detail is required here; also, the reference to a tech report does not provide strong support

"If there is interval uncertainty ... . Then" - > "If there is interval uncertainty ... , then"

Section 3

There is no discussion of the optimality or runtime of the algorithms that are presented.

"The first is to consider that the traditional definitions of sensitivity and
specificity can be could be re-imagined by defining what the predictive sensitivity" - grammar

"for characterising the uncertainty with . Their"

"therefore constructs consider"

It seems the methods by de Souza and Billard-Diday could be trivially adapted to output intervals by simply not taking the mean. How does that compare to the proposed approach?

Section 4

"to find an esitmate"

**Strengths And Weaknesses:**


Learning a logistic regression model from interval-valued data appears to be a subproblem considered in the area of symbolic data analysis, see, for example,

Beranger, B., Lin, H., & Sisson, S. (2022). New models for symbolic data analysis. Advances in Data Analysis and Classification, 1-41.

but the submission fails to present it in this context.

There is also relevant work that is not discussed or compared to:

de Barros, A. P., de Carvalho, F. D. A. T., & Neto, E. D. A. L. (2012, October). A pattern classifier for interval-valued data based on multinomial logistic regression model. In 2012 IEEE International Conference on Systems, Man, and Cybernetics (SMC) (pp. 541-546). IEEE.

Whitaker, T., Beranger, B., & Sisson, S. A. (2021). Logistic regression models for aggregated data. Journal of Computational and Graphical Statistics, 30(4), 1049-1067.

Whitaker, T. (2019). Innovative methods for the analysis of complex and non-standard data (Doctoral dissertation, UNSW Sydney).

Moreover, according to

Pełka, M., & Rybicka, A. (2019). Identification of factors that can cause mobile phone customer churn with application of symbolic interval-valued logistic regression and conjoint analysis. In The 13th Professor Aleksander Zelias International Conference on Modelling and Forecasting of Socio-Economic Phenomena (pp. 187-195).

the reference de Souza (2011) actually discussed three different methods for using interval-valued data in logistic regression, not just the one method discussed in the submission.

The submission should also discuss

Fagundes, R. A., De Souza, R. M., & Cysneiros, F. J. A. (2014). Interval kernel regression. Neurocomputing, 128, 371-388.

and explain why this cannot be applied to kernel logistic regression.

Other weak points of the submission:

- Only one real-world dataset with interval-valued data is used for the experiments (intervals are introduced artificially in the other datasets).

- The algorithms proposed are either brute-force and not computationally feasible in realistic scenarios or apply heuristics without any performance guarantees.

- Two of the competitors considered in the study can be trivially adapted to output an interval rather than a single estimate based on the mean.

---

### Review · Reviewer_9TdV · 2022-10-15

**Summary Of Contributions:**

The paper explores the probabilistic nature of classifiers, namely, of the logistic regression. It is proposed to estimated a number of logistic regression models to construct interval uncertainties.


**Requested Changes:**

The algorithm lacks theoretical foundations, however, if it could be shown that it performs well in practice (i.e., add much more results of numerical experiments on realistic data and compare them to the state-of-the-art), it would be interesting.


**Strengths And Weaknesses:**

Strengths. The method is described in details. The evaluation metrics are also well explained.


Weaknesses.
I still do not understand why \beta should be minimised or maximised, if it is the function which is supposed to be minimised/maximised.
(I reviewed the previous version of the paper).
In de Souza method "present a method for characterising the uncertainty with ." Please finish the sentence.

The paper was carefully re-written compared to its previous version. However, in general, the description and explanations of the results are still quite incremental and straightforward. The approach seems to be a heuristic, since there is a number of claims without any proofs.

It is not discussed in details in the current submission but I suppose that the model is rather complex and I wonder whether it is applicable even to a moderate size task. Also Section 4 introduces a very complex model with an exponential number of logistic regressions (2^q logistic regression models), and the motivation for such a complex model is lacking.

There is also a lack of numerical experiments on realistic data sets.

I also wonder whether bootstrap can be used as a baseline method for the proposed approach.

---

### Review · Reviewer_Sxmf · 2022-10-19

**Summary Of Contributions:**

The paper proposes a new method to handle interval data both at the input and at the output of logistic regression. The idea of the approach is to construct the set of all possible logistic regression models that can be learned when input and output values are drawn from these intervals and then to provide predictions in the form of intervals as well (constructed from all possible models). Experiments are conducted on several artificial and real datasets.


**Broader Impact Concerns:**

I have no concern about the ethical implications of the work.

**Requested Changes:**

All the issues raised above require changes in the paper. If (1) can not be addressed, the paper could maybe refocus on uncertainty of the features only, although (2) is still a problem to me. Other points can be more easily addressed I think.

**Strengths And Weaknesses:**

The idea of constructing the set of the all possible models that can be learned from instances of the dataset compatible with the intervals is very interesting. The fact that it is possible to reduce this set using (8) is a very nice result. I appreciate the fact that exact and approximate algorithms are provided in both cases (input and output uncertainty). The paper is well written and the authors have put a lot of effort in the assessment and the visualization of their results.

The paper has several important weaknesses however at this stage.

(1) I'm not convinced by the approach in the case of label uncertainty. The problem in this setting is that adding unlabeled examples increases uncertainty with respect to not using them, and where these examples lie in the input space can have a huge impact on this uncertainty (as shown in Figure 9). In front of a new dataset with unlabeled examples, one thus has the strange choice between not using these unlabeled examples, which will give no uncertainty about the model, or using them, which will yield a important uncertainty. As, formally, every $x$ without a label can be considered as a unlabeled examples, one can thus arbitrarily increase uncertainty by adding new unlabeled examples, which does not seem right to me. I think that the problem comes from the fact that the authors do not assume a particular data model. Not knowing at all the label of an example and not making any assumption about the data distribution should make the unlabeled examples uninformative, and thus useless, to me.

(2) To a smaller extent, there is a similar problem in the case of input intervals. Not having interval data gives the feeling that there is no uncertainty about the model, while adding examples with uncertain inputs gives more uncertain outputs, which is contradictory to me (having more data, even uncertain, should reduce the uncertainty about the model, not increasing it). I think the problem is that the original epistemic uncertainty of logistic regression, with non interval data, is not integrated into the equation. To me, this makes the interpretation of the interval predictions provided by the model very hazardous. This problem is overlooked in the paper I think.

(3) In the case of uncertainty in features, the paper is lacking a proof that it is indeed possible to reduce the set of models to (8). Maybe, it's trivial but I was not able to convince myself that the models in (8) are enough to provide minimum and maximum values for $\pi(x)$ for all $x$. A formal proof should be given.

(4) The description of the algorithms is not detailed enough. In Algorithm 1, what means "using stochastic optimization"? How is it implemented in practice? What is the computational cost of this step? Algorithm 2 needs also more explanation. What means "many $P$"? What is $P$ actually? A point or a prediction interval? The first step of the loop makes $T$ an input to $\pi$ and $P$ an output of $\pi$ but later in the code $P$ is compared to $T$. So, I'm unable to understand Algorithm 2. It needs to be explained further in the text.

There is also not discussion about the hyper-parameters of the algorithms and how they have been set in the experiments. Experiments are no reproducible.

(3) Experimental results are nicely presented but the experiments however leave me wanting more. In both scenarios, only a single purely manually constructed dataset and a single real dataset, but with manually constructed intervals, are used to illustrate the method and compare it against competitors. Only the problem in Section 5.1 is a real dataset with interval data. This is a bit short. In addition, these experiments are purely illustrative. I'm missing for example more systematic experiments that compare Algorithm 1 with Algorithm 2, as well as Algorithm 3 with Algorithm 4, and study the impact of their hyperparameters. How well does Algorithm 2 (resp. 4) approximate the result of Algorithm 1 (resp. 3), and how much does it improve computing times?

(4) The comparison about state-of-the-art methods is not totally convincing. None of the competitors provide interval predictions and thus I don't know what conclusions can be drawn from these comparisons. But I'm not aware of other methods that takes into account interval data and provides intervals at the output. Maybe the author could propose a way to provide a pointwise prediction from their intervals (e.g., taking the midpoint of the interval) and compare quantitatively this prediction against other methods (using AUC).

(5) Some typos/errors in the text:
- Algorithm 1: $D'$ is used instead of $D'_{\beta_i}$ (twice)
- Algorithm 2: $cud_{min}$ instead of $d_{min}$
- In (9), $\in$ is missing in the third case
- Page 8, the first sentence in the description of the de Souza method is unfinished
- Page 9, "constructs consider"
- Page 9, Section 3.2: "Dataset $X$ from 2": I took me some time to undertstand that you were referring to the dataset used in Section 2.1. The use of $X$ and $Y$ as dataset names in the different figures is confusing at first. I would have preferred $D_1$ and $D_2$ for example.
- Page 11, last sentence: I would add a table with the AUC values.
- Page 13, top: "shown in a the intervals are plotted" ?
- Page 13, Section 4, "contains p variables" => "p examples"
- Pages 14: "contains points for which have" (twice)

---

### Decision · Action_Editors · 2022-12-03

**Recommendation:** Reject

**Comment:**

On top of the several significant weaknesses pointed out in most of the reviews, I am quite concerned by the lack of engagement from authors: not all reviews have been addressed by rebuttals, and several reviewers expressed their disappointment that their time and effort in reviewing has not been acknowledged and used by authors. The authors did not respond directly to each reviewer, who wrote extensive and detailed reviews, and indicate what changes were made. The paper does not reach the acceptance bar for TMLR in my opinion, and I do not recommend a resubmission.

***This meta-review was edited by the Editors in Chief to improve the explanation of the decision.

**Audience:**

This is hard to assess -- some reviewers remain unconvinced with the proposed approach so it is unclear at this stage whether this can attract some interest from the TMLR community.

The revised paper is difficult to read and has many typos and unclear formalization of the algorithms, despite detailed reviewer comments. This lack of clarity in presentation reduces the value for the TMLR community.

**Claims And Evidence:**

The paper has definitely improved compared to the original version. It is now clearer that the proposed methods are intended to be heuristic in nature. Time complexity has also been assessed now, but there are no assessments of actual computational time. This is a critical shortcoming because time complexity measures the number of steps, but the steps can take a different amount of computation within different algorithms considered in the paper (e.g. some algorithms involve optimisation).

Overall, the paper still is lacking in motivation and evidence and the reviewers do not believe that the authors have supported their claim that this method is appropriate to handle uncertainty.